# Masked Autoencoders with Multi-Window Local-Global Attention Are Better Audio Learners

**Sarthak Yadav**[1,2]   **Sergios Theodoridis**[1,4]   **Lars Kai Hansen**[2,3]   **Zheng-Hua Tan**[1,2]
[1]Aalborg University    [2]Pioneer Centre for Artificial Intelligence, Denmark
[3]Technical University of Denmark    [4]University of Athens
`[sarthaky,sthe,zt]@es.aau.dk, lkai@dtu.dk`

## Abstract

In this work, we propose a Multi-Window Masked Autoencoder (MW-MAE) fitted with a novel Multi-Window Multi-Head Attention (MW-MHA) module that facilitates the modelling of local-global interactions in every decoder transformer block through attention heads of several distinct local and global windows. Empirical results on ten downstream audio tasks show that MW-MAEs consistently outperform standard MAEs in overall performance and learn better general-purpose audio representations, along with demonstrating considerably better scaling characteristics. Investigating attention distances and entropies reveals that MW-MAE encoders learn heads with broader local and global attention. Analyzing attention head feature representations through Projection Weighted Canonical Correlation Analysis (PWCCA) shows that attention heads with the same window sizes across the decoder layers of the MW-MAE learn correlated feature representations which enables each block to independently capture local and global information, leading to a decoupled decoder feature hierarchy.

## 1 Introduction

With rapid advances in hardware capabilities driving models of ever-increasing complexity and appetite for data, leveraging unlabelled data for learning effective deep representations has garnered significant interest. Self-supervised learning, which solves a pretext task that utilizes labels generated from the data itself, has emerged as a notable approach for training deep neural representations without labelled data. Several methods for learning self-supervised representations from audio data have been proposed, including autoregressive methods that learn to predict the future based on the past input (Oord et al., 2018; Chung et al., 2019), methods that learn contrastive representations from different views of the input (Saeed et al., 2021; Schneider et al., 2019; Baevski et al., 2020; Sarkar et al., 2019), and masked predictive modelling methods that learn to predict removed portions of the input data (Devlin et al., 2019).

Together with the transformer architecture (Vaswani et al., 2017) and its successors (Dosovitskiy et al., 2021; Liu et al., 2021), *masked predictive modelling* has led to significant advances across natural language processing (NLP) (Devlin et al., 2019; Lewis et al., 2020), computer vision (Xie et al., 2022; Bao et al., 2022) and audio and speech processing (Hsu et al., 2021a). Masked Autoencoders (MAEs) by He et al. (2022) are a recent addition to the masked predictive modelling family. Initially proposed for learning visual representations from randomly masked image patches, MAEs are experiencing widespread adoption across several domains (Feichtenhofer et al., 2022; Wei et al., 2022; Hou et al., 2022; Pang et al., 2022; Bachmann et al., 2022; Seo et al., 2023) due to their inherent scalability and simple design. In the audio domain, several recent works have adapted MAEs to learn a general-purpose audio representation from spectrogram inputs (Baade et al., 2022; Niizumi et al., 2022). These works address several challenges that are unique to the audio domain and exhaustively study the effect of masking strategies and other hyperparameters, providing vital information for training MAEs on audio data.

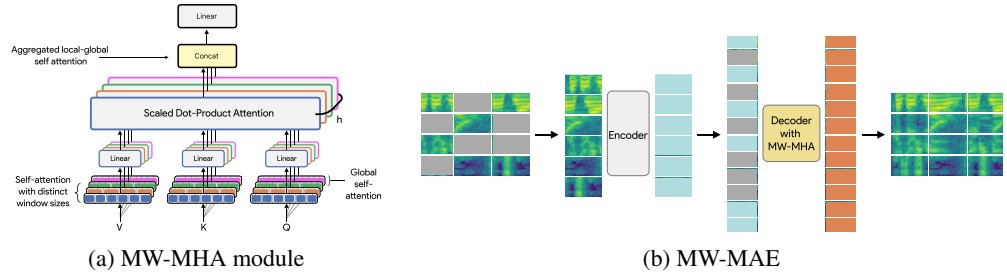

(a) MW-MHA module          (b) MW-MAE

Figure 1: An overview of the proposed Multi-Window Multi-Head Attention (MW-MHA) module, and the overall MW-MAE architecture. In MW-MHA, each attention head operates on non-overlapping windows of different sizes (coded by different colours) of the input matrices. As evident from b) MW-MAE uses the proposed MW-MHA block only in the decoder.

Recent works have shown that leveraging local information in the Multi-Head Attention (MHA) module of a transformer layer through convolutions (Gulati et al., 2020), attention with local windows (Beltagy et al., 2020; Liu et al., 2021; Dong et al., 2022) or pooling attention (Fan et al., 2021; Li et al., 2022; Zhu & Omar, 2023) can lead to improved performance. Within the framework of Masked Autoencoders, Huang et al. (2022) evaluated the impact of local windowed attention (Liu et al., 2021) for audio representation learning, demonstrating better performance across 4 downstream audio recognition tasks. However, in these methods: (i) all the attention heads within a MHA module operate at the same local context, thus only capturing local information at the transformer layer level, and (ii) they require explicit approaches to mitigate the lack of connections across windows and to capture local-global information.

In this work, we propose Multi-Window Masked Autoencoders (MW-MAE) with both local and global attention for learning general-purpose audio representations from spectrogram inputs. Decoders in an MW-MAE are fitted with a novel Multi-Window Multi-Head Attention module (MW-MHA) (Fig 1). Each attention head in the proposed MW-MHA module computes self-attention over non-overlapping windows of different sizes, which facilitates modelling of local-global interactions in every decoder transformer block. The proposed MW-MAEs outperform standard MAEs on 10 downstream audio recognition tasks. At the same time, MW-MAEs adapt better to varying patch sizes and increasing number of patches, perform better in low-data scenarios, as well as demonstrate better performance and scaling characteristics with respect to changing encoder and decoder complexities. Exhaustive exploratory analysis of attention distances and entropies shows that attention heads in MW-MAE encoders learn broader local-global attention as compared to standard MAEs, despite having an identical architecture. Further, analysis of feature representations from the decoder attention heads using Projection Weighted Canonical Correlation Analysis (PWCCA) (Morcos et al., 2018) indicates that attention heads with the same window sizes across the decoder layers of the MW-MAE learn correlated feature representations leading to a decoupled feature hierarchy, confirming that MW-MHA modules learn local-global features in each decoder block.

## 2 BACKGROUND AND RELATED WORKS

Recently, several works have been proposed for learning audio representations in a self-supervised manner. Most of these works can be loosely categorized into one or more of the following groups based on their underlying pretext task: (i) predictive; (ii) contrastive; and (iii) masked predictive modelling. Several methods adopt a predictive coding approach, which aligns well with the sequential nature of audio input. Autoregressive predictive coding (APC) (Chung et al., 2019; 2020; Chung & Glass, 2020) is one such method which utilizes Recurrent Neural Networks (RNNs) to predict future elements of a sequential input given the past, whereas non-autoregressive approaches using Convolutional Neural Networks (CNNs) have also been proposed (Liu et al., 2021). Contrastive predictive coding (Oord et al., 2018) optimizes a predictive coding objective in the latent space while simultaneously optimizing a contrastive objective function. This brings us to contrastive representation learning, which operates on the premise of learning a representation space that maximizes agreement between views from the same input sample while minimizing inter sample agreement. Several contrastive learning-based methods, originally proposed for computer vision (Chen et al.,

2020; Chen & He, 2021; Grill et al., 2020), have been adapted for learning audio representations (Saeed et al., 2021; Niizumi et al., 2021; Elbanna et al., 2022). A widely used contrastive approach for learning speech representations is the Wav2Vec family of algorithms (Schneider et al., 2019; Baevski et al., 2019; Baevski et al., 2020; Hsu et al., 2021b), which learn contextualized speech representations by optimizing a contrastive objective between quantized latent representations and representations generated from masked time steps. Finally, self-supervised learning methods based on masked predictive modelling have a simple premise: remove a portion of the input data, and learn to predict the removed content. After being (re-)popularized by the likes of BERT (Devlin et al., 2019) in NLP and fueled by the recognition of the Transformer (Vaswani et al., 2017) as a viable cross-domain neural architecture, masked modelling has seen wide adoption in several domains, such as computer vision (Bao et al., 2022; Xie et al., 2022), audio and speech (Liu et al., 2020) as well as a multi-domain self-supervised learning frameworks (Baevski et al., 2022). In the audio domain, several recent methods use masked predictive modelling to learn self-supervised representations (Baevski et al., 2019). These include HuBERT (Hsu et al., 2021a), which trains a BERT-like model to predict pre-determined cluster assignments from masked speech features, WavLM (Chen et al., 2022b), which learns a joint denoising and masked prediction task, and SSaST (Gong et al., 2022), which jointly reconstructs and contrasts masked patches. More recently, Chen et al. (2023) proposed an iterative masked modelling approach using an iterative self-distilled tokenizer that generates refined discrete labels from audio input data for the next stage of pretraining.

**Masked Autoencoders (MAEs):** Recently, He et al. (2022) proposed Masked Autoencoders for learning self-supervised image representations. In an MAE, the input is split into non-overlapping patches, which are then linearly projected to a fixed dimension by the patch embedding layer. A large subset of these patches is masked out (*e.g.*, over 75% of the patches), and the unmasked patches are then encoded by a Vision Transformer (ViT) (Dosovitskiy et al., 2021). With learnable mask tokens filled in the correct positions to restore the original patch order, these encoded patches are fed to a transformer based decoder, whose objective is to learn to reconstruct the masked patches. The high masking ratio allows large encoders to be paired with significantly smaller decoders due to the reduced encoding complexity, while simultaneously forcing the encoder to learn better contextualized representations by minimizing extrapolation from redundant neighbouring patches. Several recent works based on MAEs have been proposed for training general-purpose audio representations. Baade et al. (2022) explored a joint discriminative and generative objective for training audio MAEs and evaluated fine-tuning performance on 5 downstream tasks. By training shallow downstream classifiers on 15 downstream tasks in accordance with the HEAR-2021 (Turian et al., 2022) protocol, Niizumi et al. (2022) investigated various hyper-parameters such as patch size and the effect of input audio clip duration on model performance. Huang et al. (2022) investigated shifting windows based local self-attention (Liu et al., 2021) of a fixed context ($4 \times 4$ windows) in all but the last few layers of audio MAEs. In contrast, in this work we propose a Masked Autoencoder fitted with a novel Multi-Window Multi-Head Attention module that can model attention at several context levels and can capture local-global interactions in every transformer layer.

## 3 PROPOSED APPROACH

### 3.1 MULTI-WINDOW MULTI-HEAD ATTENTION

To better capture local-global attention, we propose a Multi-Window Multi-Head Attention (MW-MHA) module, where each attention head computes self-attention across spectrogram patches in different local windows and then combines the contribution of each attention head, as illustrated in Figure 1a. We define MW-MHA with $h$ parallel heads as follows:

$$\text{MWMHA}(Q, K, V) = \text{Concat}(\text{winHead}_1, ..., \text{winHead}_\text{h})W^O \tag{1}$$

$$\text{winHead}_i = \text{WinAttention}(QW_i^Q, KW_i^K, VW_i^V, win_i) \tag{2}$$

Here, $Q, K, V \in \mathbb{R}^{n \times d_m}$ represent query, key and value input matrices, and $W_i^Q, W_i^K, W_i^V \in \mathbb{R}^{d_m \times d_k}$ are their corresponding learnable projection matrices. $d_m$ is the model's feature dimension, and $d_k = \frac{d_m}{h}$. As opposed to MHA, each head $\text{winHead}_i$ computes local self-attention over non-overlapping windows of size $win_i$ by partitioning input matrices $QW_i^Q, KW_i^K, VW_i^V \in \mathbb{R}^{n \times d_k}$

into $Q_{win_i}, K_{win_i}, V_{win_i} \in \mathbb{R}^{m \times win_i \times d_k}$, given that $n = m \times win_i$. This is followed by computing standard self-attention $X_{win_i} = \text{Attention}(Q_{win_i}, K_{win_i}, V_{win_i})$ (Vaswani et al., 2017) on these partitioned inputs. Finally, $X_{win_i} \in \mathbb{R}^{m \times win_i \times d_k}$ is reshaped to $X \in \mathbb{R}^{n \times d_k}$ to get the output.

In the proposed MW-MHA module, individual attention heads capture information at multiple local contexts, and the final projection matrix $W_i^O \in \mathbb{R}^{hd_k \times d_m}$ learns the contribution of each of these heads, allowing inter-window interaction and connection. This design facilitates learning both *local* and *global* time-frequency information at several granularities in every transformer block (as supported by exploratory analysis in Section 5). This is in contrast to shifting (Liu et al., 2021; Chen et al., 2022a), striped (Dong et al., 2022) windowed self-attention, or pooling attention (Fan et al., 2021; Li et al., 2022; Zhu & Omar, 2023), where all attention heads within the same block have the same window size and thus only perform local self-attention at the block level. Pseudo-code for the proposed MW-MHA is provided in Appendix A.

### 3.2 Masked Autoencoder with Multi-Window Multi-Head Attention

**Patch embeddings, masking strategy and masking ratio:** We use mel-spectrograms as inputs, partitioning them into non-overlapping patches, which are then flattened and embedded into linear projections. For encoding positional information, we use fixed sinusoidal positional embeddings, similar to Baade et al. (2022); Niizumi et al. (2022); Huang et al. (2022). We use a high masking ratio (80%) and random unstructured masking, which have been shown to work well for audio (Niizumi et al., 2022; Huang et al., 2022).

**Encoder:** In line with previous work (He et al., 2022; Niizumi et al., 2022; Huang et al., 2022), we use a Vision Transformer (ViT) (Dosovitskiy et al., 2021) based encoder, which only processes non-masked patches (20% in this work). Due to the random masking strategy, majority of the patches are not processed by the encoder at training time. This minimizes the benefit of using the proposed MW-MHA modules in the encoder transformer blocks (as evidenced by experiments in Section 4.4). Thus, transformer blocks in our encoder use standard Multi-Head Attention.

**Decoder with Multi-Window Multi-Head Attention:** We add fixed sinusoidal positional embeddings to the encoded visible patches concatenated with trainable *masked tokens* after restoring original patch order. The resulting tensor is then fed to the decoder, which is also a stack of transformer layers, followed by a linear head that reconstructs the original input spectrogram. This is consistent with previous works (He et al., 2022; Niizumi et al., 2022; Huang et al., 2022). Given that the decoder processes all the patches, we replace the Multi-Head Attention module with the proposed Multi-Window Multi-Head Attention, thus modelling local-global attention in every decoder block.

**Selecting window sizes:** We follow a simple rule for determining the window sizes of each constituent $\text{winHead}_i$: given the total number of patches $n_p$, we simply take all non-unary factors of $n_p$ and add two additional global self-attention heads. As an example, our default configuration yields $n_p = 250$, and thus the window sizes for each MW-MHA module in all decoder blocks will be $[2, 5, 10, 25, 50, 125, 250, 250]$ for a total of 8 attention heads, which is a reasonable number of attention heads inline with previous research (He et al., 2022; Huang et al., 2022). Not only is this method simple to follow, but it also scales well with number of patches, effectively covering several possible local context levels.

**Pre-training objective:** During pre-training, we optimize a loss function that computes mean squared error (MSE) between the predicted masked patches and their corresponding input spectrogram patches. In early experiments, we observed reduced performance when using per-patch normalization, and thus we do not normalize target spectrogram patches.

## 4 Experiments

### 4.1 Datasets and Tasks

**Pre-training:** We use the full AudioSet dataset (Gemmeke et al., 2017) (AS-5k) for pre-training MAEs and MW-MAEs. With over 5000 hours of audio data distributed in 2 million 10-second weakly annotated YouTube clips spanning 527 classes, AudioSet is one of the largest publicly available audio corpora.

**Downstream tasks:** Recently, several standardized benchmarks have been proposed to evaluate audio representations thoroughly across a wide variety of domains, such as SUPERB (Yang et al., 2021a) and HEAR (Turian et al., 2022). While both benchmarks offer avenues for fast, reproducible and accessible comparison of audio representations, the SUPERB benchmark focuses primarily on speech-processing applications. In contrast, the HEAR benchmark consists of 19 tasks spanning diverse audio domains of speech, music and environmental sounds and redistributes standardized and preprocessed datasets. However, some of these tasks are simply smaller subsets of one another, whereas performance on some HEAR tasks has been demonstrated to be correlated (Turian et al., 2022). For evaluating audio representations, we utilize a subset of the HEAR benchmark which consists of ten diverse tasks spanning multiple domains: Beijing Opera, Crema-D, ESC-50, LibriCount, Mridangam Stroke and Tonic, NSynth Pitch 5h, Speech Commands 5h, FSD50K and VoxLingua107. More information can be found in Appendix B along with the underlying selection criterion. We believe the selected tasks constitute a balanced evaluation protocol that facilitates assessment of audio representations without doing excessive evaluations. For downstream evaluation, we follow the HEAR protocol, where for each task, a shallow downstream classifier is trained on top of fixed features extracted using a pretrained model. This practice has become quite prevalent and allows the evaluation of how representations generalize to a broad range of tasks without the drawbacks of fine-tuning large, heterogeneous neural networks.

**Measuring overall performance:** Given the wide variety of downstream tasks and feature representations evaluated, a single metric to quantify the performance would significantly aid analysis. However, given the differing difficulty levels of the tasks as well as outliers arising from the nature of the representations evaluated, simply averaging the scores is not sufficient. To counteract this, we utilize a normalized overall score to track overall performance of a given audio representation. Mathematically, overall score $s(m) \in [0., 100.]$ of a model $m$ is given as:

$$s(m) = \frac{1}{|T|} \sum_{t \in T} \frac{x_t(m) - min_t}{max_t - min_t} * 100 \tag{3}$$

where $x_t(m)$ denotes performance of the model $m$ on task $t$, and $min_t$ and $max_t$ represent the worst and the best performance across all models on the task. By taking the relative performance of the best and the worst approach on a task into consideration, this overall score takes *how hard the task is to improve on* in consideration. It is worth noting that the normalized score is computed across all the evaluated methods in all upcoming sections, including ablations. This is similar to the overall score used by the public leaderboard of the SUPERB Yang et al. (2021a) benchmark, except that we do not set the normalized value of the worst performing method to 0, and the proposed overall score has an upper range of 100.0.

## 4.2 IMPLEMENTATION DETAILS

**Features:** We use log-scaled mel spectrograms with a window size of 25 ms, a hop size of 10 ms and $F = 80$ mel-spaced frequency bins in the $50 - 8000$ Hz range, extracted using the *torchaudio* (Yang et al., 2021b) toolkit. All datasets have a sampling frequency of 16000 Hz. Instead of normalizing by dataset statistics, we adopt a per-instance standardization scheme.

**Pre-training:** We use the AudioSet dataset for pre-training our Masked Autoencoders. We extract log-scaled mel spectrograms for the entire AudioSet dataset and randomly crop a segment 200 timesteps in length from each data sample. Our default configuration consists of a ViT-B encoder. All our MAE variants accept a $200 \times 80$-dimensional (T $\times$ F, respectively) input corresponding to an audio duration of 2 seconds, which achieves performance on-par with longer input durations as demonstrated by Niizumi et al. (2022). For our default configuration, our patch embedding computes non-overlapping patches with a patch size of $(4 \times 16)$, given it's desirable performance v/s complexity tradeoff as found by Niizumi et al. (2022). A key characteristic of the Masked Autoencoder paradigm is its asymmetric design, which allows pairing small decoders with large encoders while scaling favourably for linear probe performance (He et al., 2022). Thus, in contrast to Huang et al. (2022), we adopt a smaller 4-layer deep transformer-based decoder of width 384 and 8 attention heads for our default configuration. We train Masked Autoencoders with the proposed MW-MHA module, which are referred to as MW-MAEs, as well as their standard MAE counterparts. All MAEs are pre-trained for 100 epochs with a batch size of 1024 and a weight decay of 0.05 on a single TPU-v3 VM with 8

Table 1: Comparison with various audio representations from the literature. $95\%$ confidence intervals are reported over 10 runs on downstream classifiers. We pre-trained all highlighted audio representations, with different gray levels indicating directly comparable MAE and MW-MAE configurations. We also pre-trained an AudioMAE model ("AudioMAE-B-4x16-4l") from scratch, which is directly comparable to our base configurations. We only pre-trained For other pre-trained audio representations, publicly available official implementations were used. All downstream models were trained by us using the *hear-eval-kit*. $s(m)$ denotes the proposed normalized overall score (Sec 4) *: same configuration as MSM-200 16x4 (Niizumi et al., 2022), with 8 attention heads in the decoder instead of 6. For model parameter counts, refer to Appendix G

| Model | PT-Data | BO | CD | ESC-50 | LC | Mri-S | Mri-T | NS-5h | SC-5h | F50K | VL | $s(m)$ |
|---|---|---|---|---|---|---|---|---|---|---|---|---|
| **Naive Baselines** | | | | | | | | | | | | |
| HEAR-Naive (Turian et al., 2022) | - | 52.6±2.4 | 30.9±0.8 | 5.8±0.2 | 33.5±1.1 | 38.0±1.3 | 36.4±1.9 | 18.6±4.4 | 8.5±0.4 | 7.1±0.2 | 11.2±0.5 | 5.0±0.7 |
| **Supervised** | | | | | | | | | | | | |
| PaSST-base (Koutini et al., 2022) | AS-5k | 94.9±0.5 | 61.0±0.3 | **94.8±0.3** | 60.1±0.2 | 96.5±0.1 | 87.6±0.6 | 23.3±0.9 | 66.6±1.4 | **64.2±0.1** | 25.5±0.8 | 73.5±0.4 |
| **SSL** | | | | | | | | | | | | |
| W2V2-base (Baevski et al., 2020) | LS-960 | 74.0±1.0 | 46.4±0.3 | 31.1±0.4 | 51.2±0.2 | 77.3±0.2 | 55.1±0.3 | 7.4±0.8 | 90.8±0.3 | 18.1±0.1 | 35.5±0.8 | 43.1±0.2 |
| W2V2-large (Baevski et al., 2020) | VP-100k | 93.1±0.7 | 66.9±0.4 | 60.1±0.5 | 62.4±0.3 | 93.9±0.1 | 77.4±0.2 | 42.0±1.0 | 87.6±0.5 | 34.2±0.1 | 53.6±1.0 | 74.0±0.4 |
| WavLM-base (Chen et al., 2022b) | LS-960 | 89.4±0.7 | 56.3±0.2 | 46.6±0.4 | 63.2±0.3 | 95.1±0.1 | 83.4±0.2 | 37.3±0.8 | 57.2±0.8 | 29.9±0.1 | 22.6±0.6 | 60.5±0.2 |
| WavLM-large (Chen et al., 2022b) | Mix-94k | 96.4±0.5 | 57.2±0.2 | 47.9±0.4 | 61.1±0.3 | 96.8±0.1 | 89.5±0.1 | 53.7±0.5 | 46.2±0.8 | 29.0±0.1 | 23.7±0.9 | 64.0±0.2 |
| HuBERT-base (Hsu et al., 2021a) | LS-960 | 92.1±0.6 | 70.8±0.2 | 57.8±0.6 | 56.5±0.3 | 94.4±0.1 | 84.9±0.3 | 19.4±0.7 | **93.2±0.1** | 32.3±0.1 | 61.8±0.6 | 72.5±0.2 |
| HuBERT-large (Hsu et al., 2021a) | LL-60k | 94.1±0.7 | 70.7±0.1 | 60.3±0.4 | 59.9±0.2 | 95.3±0.1 | 83.5±0.3 | 19.3±0.8 | 83.2±0.7 | 31.5±0.1 | **66.1±0.9** | 73.4±0.3 |
| SSaST-base (Gong et al., 2022) | AS+LS | 93.4±0.9 | 56.5±0.2 | 68.4±0.4 | 60.7±0.3 | 96.7±0.1 | 96.3±0.1 | 66.8±0.7 | 53.5±1.3 | 38.2±0.1 | 28.5±0.9 | 71.7±0.2 |
| BEATs-iter3 (Chen et al., 2023) | AS-5k | 94.0±0.8 | 67.3±0.2 | 83.7±0.3 | 68.0±0.2 | 94.7±0.1 | 95.8±0.1 | 69.4±0.8 | 85.2±0.3 | 53.6±0.2 | 38.5±1.0 | 85.7±0.3 |
| **MAE based** | | | | | | | | | | | | |
| AudioMAE (Huang et al., 2022) | AS-5k | 93.7±0.6 | 68.2±0.2 | 60.6±0.4 | 42.2±0.2 | 89.2±0.2 | 86.6±0.2 | 64.5±0.8 | 28.6±1.5 | 37.9±0.1 | 29.7±1.0 | 62.9±0.3 |
| AudioMAE-B-4x16-4l | AS-5k | 96.0±0.5 | 72.4±0.3 | 72.0±0.5 | 66.9±0.4 | 97.2±0.0 | 98.2±0.1 | 69.8±0.8 | 89.8±0.3 | 49.0±0.1 | 38.3±0.8 | 86.1±0.3 |
| MAE-B-4x16-4l* | AS-5k | 96.2±0.3 | 72.2±0.2 | 80.9±0.4 | 67.3±0.3 | 97.4±0.1 | 98.3±0.1 | 68.3±0.4 | 89.4±0.3 | 50.4±0.1 | 43.1±0.9 | 88.1±0.2 |
| MAE-B-5x5-4l | AS-5k | 96.0±0.4 | 70.9±0.2 | 80.9±0.4 | 67.6±0.4 | **97.6±0.1** | 98.4±0.0 | 69.3±0.4 | 88.4±0.3 | 49.3±0.2 | 37.7±0.6 | 86.8±0.2 |
| MAE-L-4x16-8l | AS-5k | 96.1±0.4 | 73.8±0.1 | 81.6±0.3 | 68.5±0.2 | 97.6±0.1 | 98.3±0.0 | 69.0±0.5 | 91.2±0.2 | 51.8±0.1 | 46.9±0.8 | 90.0±0.2 |
| **Proposed** | | | | | | | | | | | | |
| MW-MAE-B-4x16-4l | AS-5k | 96.0±0.5 | 73.1±0.3 | 81.2±0.4 | 68.8±0.2 | 97.4±0.1 | 97.9±0.1 | 69.3±0.6 | 90.9±0.2 | 51.2±0.2 | 44.2±0.9 | 89.2±0.2 |
| MW-MAE-B-5x5-4l | AS-5k | **96.6±0.4** | 73.8±0.4 | 82.0±0.3 | **70.1±0.4** | 97.5±0.1 | **98.3±0.1** | 72.9±0.5 | 91.7±0.2 | 51.3±0.1 | 44.2±0.6 | 90.6±0.1 |
| MW-MAE-L-4x16-8l | AS-5k | 95.9±0.3 | **76.1±0.2** | 83.6±0.3 | 69.7±0.3 | 97.4±0.0 | 98.2±0.1 | 71.2±0.7 | 93.0±0.1 | 53.5±0.1 | 51.9±0.7 | **92.6±0.2** |

TPU cores, with the default configuration taking around 36 hours to train. We warm up for ten epochs to a base learning rate of 1e-5, followed by a cosine decay schedule. A masking ratio of 0.8 with unstructured random masking is used, and no other data augmentations are used during pre-training.

**Training downstream models:** We first extract fixed feature embeddings for all downstream tasks to train downstream models. In the MAE framework, the decoder is discarded after pretraining and feature embeddings are extracted using just the encoder. To generate scene embeddings consistent with the HEAR protocol, we use the exact patch aggregation process as Niizumi et al. (2022): we break audio clips into non-overlapping 2 second chunks, concatenating the features in time and finally taking a mean over the time axis to generate a fixed vector representation independent of the input audio duration. The *hear-eval-kit*, released alongside the HEAR benchmark, was used to extract fixed feature embeddings and to train a shallow MLP classifier with a single hidden layer with 1024 neurons for each task in a reproducible manner. Experiments are repeated with at least ten random seeds for each task, resulting in 100 experiments for every evaluated representation.

## 4.3 COMPARISON WITH EXISTING WORKS

Table 1 shows how MW-MAE fares against recent audio representations. The highlighted model configurations that we pre-trained from scratch on AudioSet have the following naming convention: the first substring shows the type of MAE (vanilla or proposed MW-MAE), followed by a single alphabet denoting ViT Encoder configuration. This is followed by the patch size used, and finally, the depth of the decoder. It's worth noting that while embedding sizes of MAE and corresponding MW-MAE configurations are the same, the embedding sizes of other methods can be different. This is inline with the current consensus of evaluating self-supervised representations in the audio domain (Yang et al., 2021a; Turian et al., 2022). MW-MAE configurations outperform all other

comparable MAEs, with the largest "MW-MAE-L-4x16-8l" configuration outperforming all the methods in overall performance (92.6±0.2). MW-MAEs also outperforms AudioMAE with standard shifting window based attention, as well as the recent BEATs-iter3 approach, which is the pre-trained representation obtained after 3 stages of self-distilled learning as proposed by Chen et al. (2023). MW-MAEs perform exceptionally well on pitch perception (NS-5h), while achieving performance on-par with speech specific representations such as WavLM, HuBERT and Wav2Vec2 (denoted W2V2) for Keyword spotting (SC-5h). Perhaps more surprisingly, they outperform speech representations trained on much larger training sets on the emotion recognition (CREMA-D) as well as speaker count classification (LibriCount) tasks. While PaSST, which is a recent state-of-the-art approach for training supervised transformers on AudioSet, outperforms every model on ESC-50 and FSD50K tasks, the overall performance of the proposed approach is significantly better. Overall, the proposed MW-MAEs learn a better general-purpose audio representation than standard MAEs, generalizing well to several audio domains and demonstrating excellent overall performance in comparison to recent audio representations.

## 4.4 KEY MODEL CHARACTERISTICS

We conduct several experiments to examine key differences between MAE and the proposed MW-MAE. While we have only reported overall score $s(m)$, detailed results for all these experiments, along with additional experiments on attention heads and window sizes can be found in Appendix H.

**MW-MHA in the encoder:** As previously mentioned, adding MW-MHA to the encoder block does not improve downstream performance. Further, we also investigate the impact of linear probing as well as fine-tuning the entire encoder stack for in-domain classification on AudioSet-20k balanced subset. No data augmentations were used. As evident from Table 2, when compared with including MW-MHA blocks in the decoder only, there is no performance benefit to adding MW-MHA

| Model | Downstream $s(m)$ | Linear Probe (mAP) | Fine-tuning (mAP) |
|---|---|---|---|
| MAE Base | 88.1±0.2 | 23.1±0.0 | 26.1±0.4 |
| MW-MAE Base (decoder only) | 89.2±0.2 | 24.2±0.1 | 26.1±0.7 |
| MW-MAE Base (encoder only) | 89.1±0.3 | 23.6±0.0 | 26.1±0.3 |
| MW-MAE Base (enc+dec) | 89.1±0.3 | 24.0±0.1 | 26.2±0.0 |

Table 2: Performance impact of MW-MHA module placement

blocks to the encoder for neither downstream performance, nor for in-domain linear probe on AudioSet-20k. However, when fine-tuning the entire encoder stack, adding MW-MHA blocks to both the encoder and the decoder provides a slight improvement and is worth considering.

**Performance impact of various patch sizes:** In an MAE, the patch embedding layer generates non-overlapping patches from the input. Thus, the size of the patch governs the number of patches as well as the time-frequency resolution that the transformer layers work at, making it an important hyperparameter to investigate.

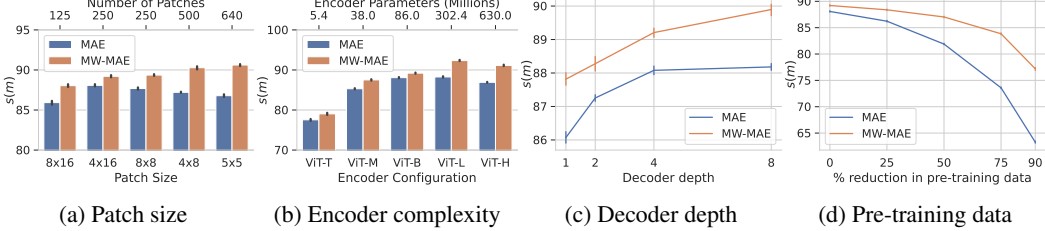

(a) Patch size     (b) Encoder complexity     (c) Decoder depth     (d) Pre-training data

Figure 2: Ablation experiments comparing standard MAE v/s proposed MW-MAE at different patch sizes (a), encoder complexity (b), decoder depth (c) as well as amount of pre-training data used (d). $s(m)$ is the proposed overall score (Sec 4). Detailed results can be found in Appendix H.

Figure 2a shows how different patch sizes affect downstream performance. The proposed MW-MAE model, with an overall score of 90.6±0.1, outperforms standard MAE for every patch size for identical decoder configurations. It's also worth noting that MAE performance degrades as we decrease the patch size beyond $4 \times 16$, whereas MW-MAE performance continues to improve. These observations show that the proposed MW-MAE adapts better to varying patch sizes and time-frequency resolutions, while scaling well with increasing number of patches.

**Encoder size:** As shown in Figure 2b, we investigate how encoders of five different complexities affect overall performance. All the trained models have the same decoder configuration (384 neurons,

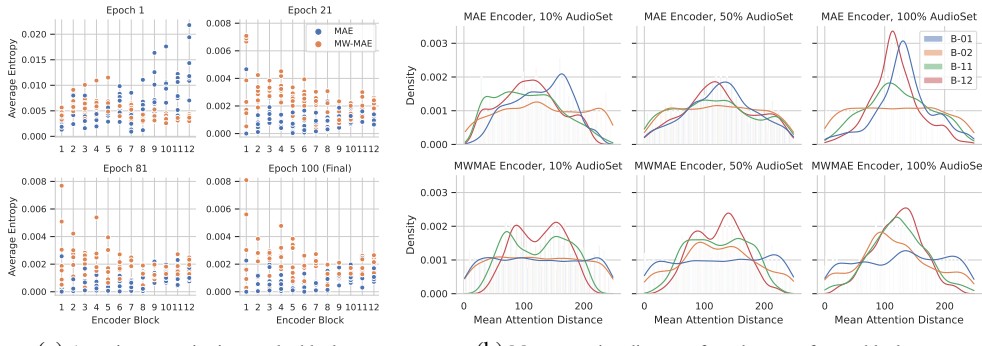

(a) Attention entropies in encoder blocks          (b) Mean attention distances for select transformer blocks.

Figure 3: Investigating MAE and MW-MAE encoder attention heads. (a) depicts average entropies of encoder attention heads over the course of pretraining in a every encoder transformer block. (b) Depicts mean attention distance distributions of the first two and the last two transformer blocks at different amounts of pretraining data used.

*depth*=4, *h*=8). With an overall score of $89.2_{\pm 0.2}$, MW-MAE with the ViT-Base encoder performs better than MAEs with encoders of any size in this experiment. The most prominent performance gap is observed for the ViT-Large setting, where MAE and MW-MAE attain overall scores of $88.2_{\pm 0.2}$ and $92.3_{\pm 0.2}$, respectively. The drop in performance for the ViT-Huge encoder for both MAEs and MW-MAEs suggests possible overfitting.

**Decoder depth:** In Figure 2c, we show how increasing decoder complexity by increasing decoder depth affects overall performance. As expected, increasing decoder depth improves performance for both methods. For decoder *depth*=8, MW-MAE ($89.9_{\pm 0.2}$) outperforms MAE ($88.2_{\pm 0.1}$) by a considerable margin in overall performance. We also observed that with an overall score of $88.3_{\pm 0.2}$, MW-MAE with *depth*=2 performs on par with MAEs with up to 4 decoder blocks. This observation complements the inherent asymmetric nature of Masked Autoencoders, and thus the proposed MW-MAE performs favourably in terms of complexity and scalability.

**Pre-training data:** Finally, Figure 2d depicts how performance varies as we reduce the amount of data used for pre-training. Overall, performance for both the MAE and the proposed MW-MAE methods continues to decrease monotonically as we remove more and more data. However, the performance loss trend for MW-MAE is much more favourable. A 90% reduction in the amount of pre-training data results in a 28.17% reduction in performance for standard MAEs (from $88.1_{\pm 0.2}$ to $63.3_{\pm 0.2}$), whereas MW-MAE only suffers a 13.5% drop in performance (from $89.2_{\pm 0.2}$ to $77.2_{\pm 0.3}$). Thus, we conclude that the proposed MW-MAEs are more adept at handling low-data scenarios in comparison to standard MAEs.

## 5 EXPLORATORY ANALYSIS

### 5.1 INSPECTING ENCODER ATTENTION HEADS

**Analyzing attention entropies:** We first analyze individual attention heads in a ViT-Medium encoder (*depth*=12, *h*=8). Figure 3a shows scatter plots of average entropies of individual encoder attention heads computed over the entire NSynth Pitch 5h validation set on a block-by-block basis at different stages during pre-training. It's worth noting that the higher the entropy, the more global the attention, with lower attention mass spent on closer tokens (Clark et al., 2019), and thus, a higher variance in entropies of individual attention heads highlights more spread out local and global attention. In the early epochs, MAE encoders actually have higher variance in entropy distribution, especially in the latter transformer layers. As pretraining goes on, interestingly, this effect is reversed, and the attention heads in the MW-MAE encoder now start converging towards high entropy variance configurations in the early layers.

**Analyzing attention distances:** We analyze mean attention distances for attention heads in the first two and the last two encoder blocks. Similar to Raghu et al. (2021), we compute attention-weighted patch distances between the query patch position and the locations it attends to for each attention head,



Figure 4: Comparing features learned by different attention heads in the encoder and the decoder of a standard MAE and the proposed MW-MAE using PWCCA. Each tick separates the attention heads of a transformer block from the next.

averaging it for all patches positions. This is repeated for all inputs in the FSD50K validation set. Figure 3b depicts the distribution of mean attention distance for MAE and MW-MAE encoders (base configuration) pretrained with different amounts of training data. We can observe that MW-MAE attention heads demonstrate a broader distribution of attention distances, modelling local-global attention better than the MAE encoder especially in the first two transformer blocks. From these observations, we can conclude that in an MW-MAE, the decoder fitted with an MW-MHA can force the encoder to better capture local-global interactions even without explicit windowed attention modules, leading to improved performance.

## 5.2 COMPARING ATTENTION FEATURE REPRESENTATIONS THROUGH PWCCA

Several recent works have used Canonical Correlation Analysis (CCA) to compare feature representations and learning dynamics of deep neural networks (Raghu et al., 2017; Pasad et al., 2021). We use Projection Weighted CCA (PWCCA) (Morcos et al., 2018), which computes a weighted mean of the CCA vectors to compare the representations learned by individual attention heads of the encoder and the decoder in identically configured MAE and MW-MAE (ViT-M encoder: *depth*=12, *h*=8; Default decoder: 384 neurons, *depth*=4, *h*=8). MW-MAE decoder uses default attention head window sizes as specified in Sec 3.2. Figure 4 depicts correlation matrices of measured PWCCA score between attention heads. We can observe a remarkable difference in correlation between the decoders: feature representations from the MW-MAE decoder attention heads with the same window sizes are strongly correlated across decoder layers, whereas attention heads with global self-attention (7, 8, 15, 16, 23, 24, 31, 32) are the least correlated, consistent with observations made for the MAE decoder. These observations suggest a decoupling of different aspects of the feature hierarchy in the MW-MAE decoder, as attention heads of specific window sizes in each decoder block capture local information at a specific granularity, which is in line with our original hypothesis. These observations are also corroborated by decoder depth ablation experiments from Sec 4.4, where we observed that a MW-MAE with a single transformer block performs on par with MAEs fitted with up to 4 decoder blocks. Finally, the difference in correlation matrices between the encoders is much less stark, which is expected since both use standard MHA blocks.

## 6 CONCLUSION

This work presents Multi-Window Masked Autoencoder (MW-MAE) for learning general-purpose audio representations. Decoders in MW-MAEs are fitted with a novel Multi-Window Multi-Head Attention (MW-MHA) module, which learns information captured at multiple granularities of local-global context by its constituent attention heads computing self-attention over different non-overlapping windows. Empirical experiments on ten downstream tasks show that the proposed MW-MAEs consistently outperform standard MAEs in overall performance when pre-trained on the AudioSet dataset, demonstrating better scaling characteristics. Exploratory analyses highlight key differences between the attention representations learned by standard MAEs and the proposed MW-MAEs. Based on attention entropy and mean attention distance analysis, we discover that encoder attention heads in an MW-MAE better capture local-global interactions, even without explicit local-global attention modules. We also learn that attention heads of the same window size across the transformer blocks of the MW-MAE decoder are correlated, learning a decoupled feature hierarchy allowing transformers to capture relevant information at the block level, supporting our original motivation.

ACKNOWLEDGEMENTS

This project is supported by the Pioneer Centre for Artificial Intelligence, Denmark. We are also grateful to the TPU Research Cloud Program, a Google Research Initiative, for providing TPU v2 and v3 devices used in this project, as well as DeiC for enabling access to the LUMI supercomputer.

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

APPENDIX

## A   MULTI-WINDOW MULTI-HEAD ATTENTION

```
def WinAttention(Q, K, V, win_i):
 n, d_k = Q.shape[-2:]
 # partition inputs along patch dimension
 # into non-overlapping windows
 Q = Q.reshape(-1, win_i, d_k)
 K = K.reshape(-1, win_i, d_k)
 V = V.reshape(-1, win_i, d_k)
 # compute self-attention
 X = softmax(Q.(K.transpose()) / sqrt(d_k)).V
 # reshape results
 X = X.reshape(-1, n, d_k)
 return X
```

Figure 5: Pseudocode for WinAttention

## B   MORE ABOUT DOWNSTREAM TASKS

Table 3: Overview of tasks for downstream evaluation. All these tasks are a part of the HEAR (Turian et al., 2022) benchmark.

| Short Hand | Name | Description | Size (in Hours) | Metric |
|---|---|---|---|---|
| BO | Beijing Opera (Tian et al., 2014; Turian et al., 2022) | Classifying percussion instruments | 0.3 | Accuracy |
| CD | Crema-D (Cao et al., 2014) | Emotion Recognition | $\sim 10$ | Accuracy |
| ESC-50 | ESC-50 (Piczak, 2015) | Environmental Sound Classification | 2.77 | Accuracy |
| LC | LibriCount (Stöter et al., 2018; Stöter et al., 2018) | Speaker Count Identification, Simulated Cocktail Party | $\sim 8$ | Accuracy |
| Mri-S | Mridangam Stroke (Anantapadmanabhan et al., 2013) | Stroke classification in pitched percussion instruments | 1.57 | Accuracy |
| Mri-T | Mridangam Tonic (Anantapadmanabhan et al., 2013) | Tonic classification in pitched percussion instruments | 1.57 | Accuracy |
| NS-5h | NSynth Pitch 5h (Turian et al., 2022; Engel et al., 2017) | 88-way Pitch Classification, reduced training subset | $\sim 5.5$ | Accuracy |
| SC-5h | Speech Commands 5h (Turian et al., 2022; Warden, 2018) | Keyword Spotting, reduced training subset | $\sim 6.5$ | Accuracy |
| F50K | FSD50K (Fonseca et al., 2021) | Multilabel, large scale Audio Tagging | $\sim 100$ | mAP |
| VL | VoxLingua107 Top10 (Turian et al., 2022; Kim et al., 2018) | Spoken language identification | 5 | Accuracy |

The following is our reasoning behind excluding the other tasks from the HEAR benchmark suite:

1. **Nsynth-Pitch 50hr and Speech Commands Full** because we already use the smaller subsets.

2. **Gunshot Triangulation:** Gunshot is an event in both AudioSet and FSD50k ontology, and is thus redundant.

3. **GTZAN Music Speech:** FSD50k already has music and speech labels, and the model performance correlation study in the HEAR paper (Turian et al., 2022) shows high correlation with FSD50k.

4. **GTZAN Genre:** highly correlated results with FSD50K and ESC-50 (surprisingly) as per (Turian et al., 2022)

5. **Vocal Imitations:** high correlation with LibriCount (Turian et al., 2022).

6. **Bee Hive state Classification:** large runtime costs, niche task.

7. **MAESTRO 5hr and DCASE 2016 Task 2:** significant complexity (storage, runtime, timestep based evaluation).

## C   EXPERIMENTAL DETAILS AND HYPERPARAMETERS

In this section, we provide additional experimental details. Apart from AudioSet, all other datasets are obtained directly from the HEAR [1], where they are pre-processed to 16000 Hz and distributed in a standard format.

Similar to Huang et al. (2022), our effective learning rate ($lr_{\text{eff}}$) depends on the base learning rate ($lr_{\text{base}}$) and the batch size as follows: $lr_{\text{eff}} = lr_{\text{base}} * \frac{\text{batch size}}{256}$. In early experiments, we did

---

[1]https://hearbenchmark.com/hear-tasks.html

not find strong augmentations at pre-training time to improve downstream performance, hence no augmentations are used. For more details, refer to Table 4. As previously mentioned, hear-eval-kit[2] was used for downstream experiments, and along with the details provided here should allow for consistent, reproducible downstream experimentation.

Table 4: **Pre-training (PT) and Downstream (FT) hyperparameters**. [*]: For ViT-L and ViT-H based models, smallest batch size that didn't give OOM was used.

| Configuration | AS-5k Pre-training | Downstream |
|---|---|---|
| Optimizer | AdamW | Adam |
| Optimizer momentum | $\beta_1 = 0.9, \beta_2 = 0.999$ | $\beta_1 = 0.9, \beta_2 = 0.95$ |
| Weight decay | 0.05 | N/A |
| Base learning rate | 0.000015 | 0.0001 |
| Learning rate schedule | linear-warmup + cosine decay | fixed |
| Minimum learning rate | 0.0 | 0.0001 |
| Dropout | 0. | 0.25 |
| Warm-up epochs | 10 | N/A |
| Epochs | 100 | 500 |
| Early Stopping | N/A | 20 |
| Batch size | 1024[*] | 1024 |
| Accelerators | 8x TPU-v3 cores | 1 Nvidia-A40 |

The code for feature extraction and running downstream experiments for our default configurations as well as the corresponding pre-trained weights can be found at `https://github.com/SarthakYadav/mwmae-jax-official`.

## D   AUXILIARY MODALITY TESTED: IMAGENET

Given the in-depth ablations and exploratory analysis, as well as resource constraints, we couldn't dive deeper into full scale testing of an additional modality given. However, as a proof of concept, we pre-trained MAE and MW-MAEs with ViT-Medium encoder on ImageNet for 100 epochs, followed by evaluating linear probe performance (training for 50 epochs) on the ImageNet validation set.

Table 5: Proof of concept ImageNet experiments.

| Model | Params | Validation Accuracy |
|---|---|---|
| MAE Medium Encoder | 38 M | $29.0_{\pm 0.0}$ |
| MW-MAE Medium Encoder | 38 M | $29.8_{\pm 0.0}$ |

## E   AUXILIARY TASK TESTED: SUPERVISED ASR

We have conducted some preliminary experiments to investigate the applicability of the proposed MW-MHA module beyond MAEs. We evaluated Supervised Transformer based ASR models, trained using CTC objective function. We trained on LibriSpeech 100h subset and reported Word Error Rates (WER) on the test sets, using validated recipes from the SpeechBrain toolkit [3]. The network architecture used consists of a 2-layer Convolutional frontend, followed by a 12-layer encoder and 4-layer decoder, each with 4 attention heads. We trained on mel-filterbanks with 80 mel bins.

Table 6: Preliminary experiments on LibriSpeech ASR.

| Model | Params | Test-Clean (WER) | Test-Other (WER) |
|---|---|---|---|
| Transformer | 8.7 M | 7.56 | 17.90 |
| MW-MHA Transformer | 8.7 M | 6.73 | 17.20 |

[2]`https://github.com/hearbenchmark/hear-eval-kit`
[3]`https://speechbrain.github.io`

For the MW-MHA Transformer, we only replaced the MHA blocks in the encoder with MW-MHA blocks, and used attention window sizes of 1/8, 1/4, 1/2 and global attention corresponding to the number of timeframes obtained from the largest audio segment in the corpus. MW-MHA transformer outperforms baseline transformer by a considerable margin on both test sets, and although we haven't yet conducted full scale librispeech experiments, this shows potential avenues for further applications.

# F ADDITIONAL EVALUATION ON A FRAME LEVEL TASK: MAESTRO-5H

We have compared our base MAE and MW-MAE configurations on the MAESTRO-5h task, which is a part of the HEAR protocol, to include proof that the proposed approach also works for frame level tasks. MAESTRO-5h is evaluated in a 5-fold cross validation setting, with Onset FMS evaluation metric. We conducted the experiments 5 times and report 95% confidence intervals over the cross validation scores of individual runs.

Table 7: Downstream performance evaluation on the MAESTRO-5h task.

| Model | Onset FMS |
|---|---|
| MAE Base | 46.47±0.05 |
| MW-MAE Base | 47.31±0.07 |

MW-MAE does demonstrate improved performance on MAESTRO-5h. While this might not look like much, at the time of writing, MW-MAE outperforms every method on the HEAR leaderboard[4] for MAESTRO-5h, including large multi and cross-modal ensemble methods, so this is excellent performance for a single method.

As previously mentioned in Appendix B, we didn't evaluate all methods on MAESTRO to begin with because it poses serious computation (and space) complexity: it takes longer to evaluate downstream performance on just MAESTRO than it takes to evaluate all the other downstream tasks combined. This was unfeasible given the number of experiments we did in this paper, more so given the fact that we evaluate on FSD50K, where local information is known to be of importance.

# G PARAMETER COUNT, AVERAGES AND OVERALL SCORES

Table 8: Models, number of parameters, plain average scores and overall scores of models omitted from Table 1 due to space constraints.

| Model | # Params | Average | $s(m)$ |
|---|---|---|---|
| HEAR-Naive | - | 24.3±0.5 | 5.0±0.7 |
| PaSST-base | 86 M | 67.5±0.3 | 73.5±0.4 |
| **SSL** | | | |
| Wav2Vec2-base | 94.4 M | 48.7±0.1 | 43.1±0.2 |
| Wav2Vec2-large | 315.4 M | 67.1±0.2 | 74.0±0.4 |
| WavLM-base | 94.4 M | 58.1±0.1 | 60.5±0.2 |
| WavLM-large | 315.4 M | 60.1±0.1 | 64.0±0.2 |
| HuBERT-base | 94.4 M | 66.3±0.1 | 72.5±0.2 |
| HuBERT-large | 315.4 M | 66.4±0.2 | 73.4±0.3 |
| SSaST-base | 89 M | 65.9±0.1 | 71.7±0.2 |
| BEATs-Iter3 | 90 M | 75.0±0.2 | 85.7±0.3 |
| **MAE based** | | | |
| AudioMAE | 86.0 M | 60.1±0.2 | 62.9±0.3 |
| MAE-B-4x16-4l | 86.0 M | 76.4±0.1 | 88.1±0.2 |
| MAE-B-5x5-4l | 86.0 M | 75.6±0.1 | 86.8±0.2 |
| MAE-L-4x16-8l | 302.4 M | 77.5±0.1 | 90.0±0.2 |
| **Proposed** | | | |
| MW-MAE-B-4x16-4l | 86.0 M | 77.0±0.1 | 89.2±0.2 |
| MW-MAE-B-5x5-4l | 86.0 M | 77.8±0.1 | 90.6±0.1 |
| MW-MAE-L-4x16-8l | 302.4 M | 79.1±0.1 | 92.6±0.2 |

---

[4]https://hearbenchmark.com/hear-leaderboard.html

# H  DETAILED ABLATION RESULTS

Tables 9, 10, 11, 12 present expanded results of our ablation experiments from Sec 4.4.

Table 9: Results from Patch size ablation experiments. ViT-B encoder was used for all experiments. $n$ denotes total number of patches, and $h$ denotes the number of attention heads in each decoder transformer block.

| Model | BO | CD | ESC-50 | LC | Mri-S | Mri-T | NS-5h | SC-5h | F50K | VL | $s(m)$ |
|---|---|---|---|---|---|---|---|---|---|---|---|
| **Patch Size=(8×16), $n$=125, $h$=4** | | | | | | | | | | | |
| MAE | 94.9±0.8 | 70.2±0.3 | 80.4±0.5 | 66.0±0.3 | 97.4±0.1 | 97.7±0.1 | 65.9±0.7 | 88.9±0.5 | 49.4±0.1 | 40.6±0.5 | 85.9±0.3 |
| MW-MAE | 95.9±0.5 | 72.3±0.2 | 81.2±0.3 | 68.4±0.3 | 97.3±0.1 | 97.8±0.1 | 67.4±0.8 | 90.0±0.3 | 50.8±0.1 | 41.9±0.5 | 88.0±0.2 |
| **Patch Size=(4×16), $n$=250, $h$=8** | | | | | | | | | | | |
| MAE | 96.2±0.3 | 72.2±0.2 | 80.9±0.4 | 67.3±0.3 | 97.4±0.1 | 98.3±0.1 | 68.3±0.4 | 89.4±0.3 | 50.4±0.1 | 43.1±0.9 | 88.1±0.2 |
| MW-MAE | 96.0±0.5 | 73.1±0.3 | 81.2±0.4 | 68.8±0.2 | 97.4±0.1 | 97.9±0.1 | 69.3±0.6 | 90.9±0.2 | 51.2±0.2 | 44.2±0.9 | 89.2±0.2 |
| **Patch Size=(8×8), $n$=250, $h$=8** | | | | | | | | | | | |
| MAE | 96.1±0.6 | 72.5±0.2 | 81.3±0.2 | 66.0±0.3 | 97.5±0.1 | 98.1±1.0 | 68.5±0.7 | 89.5±0.4 | 50.2±0.1 | 42.3±0.5 | 87.7±0.2 |
| MW-MAE | 96.3±0.4 | 73.0±0.1 | 82.6±0.3 | 69.3±0.3 | 97.5±0.1 | 98.1±0.1 | 70.3±0.8 | 90.5±0.1 | 51.4±0.1 | 42.3±0.5 | 89.4±0.1 |
| **Patch Size=(4×8), $n$=500, $h$=12** | | | | | | | | | | | |
| MAE | 96.7±0.2 | 71.3±0.3 | 79.0±0.4 | 67.8±0.3 | 97.7±0.0 | 98.5±0.0 | 68.7±0.4 | 89.0±0.4 | 49.8±0.2 | 39.2±0.7 | 87.2±0.1 |
| MW-MAE | 95.6±0.7 | 74.1±0.2 | 81.9±0.3 | 70.1±0.3 | 97.6±0.1 | 98.2±0.1 | 72.0±0.7 | 91.2±0.3 | 51.6±0.1 | 44.0±0.8 | 90.3±0.2 |
| **Patch Size=(5×5), $n$=640, $h$=16** | | | | | | | | | | | |
| MAE | 96.0±0.4 | 70.9±0.2 | 80.9±0.4 | 67.6±0.4 | 97.6±0.1 | 98.4±0.0 | 69.3±0.4 | 88.4±0.3 | 49.3±0.2 | 37.7±0.6 | 86.8±0.2 |
| MW-MAE | 96.6±0.4 | 73.8±0.4 | 82.0±0.3 | 70.1±0.4 | 97.5±0.1 | 98.3±0.1 | 72.9±0.5 | 91.7±0.2 | 51.3±0.1 | 44.2±0.6 | 90.6±0.1 |

Table 10: Effect of encoder size on performance. Patch size of 4×16 was used for all experiments.

| Model | BO | CD | ESC-50 | LC | Mri-S | Mri-T | NS-5h | SC-5h | F50K | VL | $s(m)$ |
|---|---|---|---|---|---|---|---|---|---|---|---|
| **Encoder=ViT-T** | | | | | | | | | | | |
| MAE | 95.6±0.5 | 63.2±0.2 | 70.1±0.5 | 64.6±0.3 | 97.1±0.1 | 97.4±0.1 | 66.4±0.7 | 74.3±0.8 | 41.6±0.1 | 26.4±0.6 | 77.6±0.3 |
| MW-MAE | 93.3±1.0 | 64.4±0.2 | 71.9±0.5 | 65.5±0.3 | 97.1±0.1 | 97.6±0.1 | 68.1±0.4 | 77.0±0.6 | 43.4±0.1 | 28.6±1.1 | 79.0±0.3 |
| **Encoder=ViT-M** | | | | | | | | | | | |
| MAE | 95.2±0.7 | 69.5±0.2 | 77.8±0.3 | 67.4±0.3 | 97.4±0.0 | 98.0±0.1 | 66.6±0.7 | 88.0±0.4 | 48.1±0.1 | 38.3±0.8 | 85.3±0.2 |
| MW-MAE | 95.9±0.3 | 71.8±0.3 | 80.3±0.4 | 69.7±0.1 | 97.2±0.1 | 97.8±0.1 | 68.1±0.5 | 88.8±0.6 | 49.6±0.1 | 39.8±0.8 | 87.5±0.2 |
| **Encoder=ViT-B** | | | | | | | | | | | |
| MAE | 96.2±0.3 | 72.2±0.2 | 80.9±0.4 | 67.3±0.3 | 97.4±0.1 | 98.3±0.1 | 68.3±0.4 | 89.4±0.3 | 50.4±0.1 | 43.1±0.9 | 88.1±0.2 |
| MW-MAE | 96.0±0.5 | 73.1±0.3 | 81.2±0.4 | 68.8±0.2 | 97.4±0.1 | 97.9±0.1 | 69.3±0.6 | 90.9±0.2 | 51.2±0.2 | 44.2±0.9 | 89.2±0.2 |
| **Encoder=ViT-L** | | | | | | | | | | | |
| MAE | 95.8±0.6 | 72.4±0.1 | 79.7±0.3 | 66.8±0.4 | 97.5±0.1 | 98.2±0.1 | 69.5±0.6 | 90.9±0.2 | 50.7±0.1 | 43.6±0.4 | 88.3±0.2 |
| MW-MAE | 95.7±0.5 | 75.5±0.2 | 82.5±0.5 | 70.1±0.3 | 97.4±0.0 | 98.1±0.1 | 70.7±0.6 | 93.2±0.1 | 53.3±0.1 | 51.9±0.8 | 92.3±0.2 |
| **Encoder=ViT-H** | | | | | | | | | | | |
| MAE | 96.8±0.2 | 71.1±0.2 | 78.3±0.4 | 67.1±0.2 | 97.5±0.0 | 98.5±0.0 | 67.6±0.6 | 89.6±0.1 | 49.5±0.2 | 40.0±0.7 | 86.9±0.1 |
| MW-MAE | 96.8±0.2 | 74.8±0.1 | 81.6±0.4 | 69.5±0.4 | 97.4±0.0 | 98.2±0.1 | 70.8±0.5 | 92.4±0.2 | 52.1±0.1 | 47.5±0.6 | 91.1±0.2 |

Table 11: Effect of decoder depth on downstream performance. ViT-B encoder, patch size of 4×16 were used for each experiment.

| Model | BO | CD | ESC-50 | LC | Mri-S | Mri-T | NS-5h | SC-5h | F50K | VL | $s(m)$ |
|---|---|---|---|---|---|---|---|---|---|---|---|
| *depth*=1 | | | | | | | | | | | |
| MAE | 96.4±0.2 | 69.8±0.3 | 78.9±0.3 | 67.4±0.3 | 97.4±0.1 | 97.9±0.1 | 66.4±0.8 | 88.5±0.2 | 49.4±0.2 | 39.0±1.1 | 86.1±0.2 |
| MW-MAE | 96.6±0.5 | 72.4±0.2 | 79.0±0.4 | 68.7±0.3 | 97.5±0.1 | 98.0±0.1 | 68.8±0.5 | 90.2±0.3 | 50.6±0.1 | 39.1±0.8 | 87.8±0.2 |
| *depth*=2 | | | | | | | | | | | |
| MAE | 96.8±0.3 | 71.3±0.3 | 78.8±0.2 | 68.8±0.2 | 97.4±0.1 | 98.2±0.0 | 67.2±0.6 | 90.0±0.2 | 49.6±0.2 | 39.4±0.7 | 87.3±0.1 |
| MW-MAE | 96.0±0.7 | 73.1±0.2 | 79.4±0.3 | 69.2±0.3 | 97.4±0.1 | 98.2±0.1 | 69.0±0.6 | 90.6±0.2 | 50.7±0.2 | 40.1±0.6 | 88.3±0.3 |
| *depth*=4 | | | | | | | | | | | |
| MAE | 96.2±0.3 | 72.2±0.2 | 80.9±0.4 | 67.3±0.3 | 97.4±0.1 | 98.3±0.1 | 68.3±0.4 | 89.4±0.3 | 50.4±0.1 | 43.1±0.9 | 88.1±0.2 |
| MW-MAE | 96.0±0.5 | 73.1±0.3 | 81.2±0.4 | 68.8±0.2 | 97.4±0.1 | 97.9±0.1 | 69.3±0.6 | 90.9±0.2 | 51.2±0.2 | 44.2±0.9 | 89.2±0.2 |
| *depth*=8 | | | | | | | | | | | |
| MAE | 96.3±0.3 | 71.7±0.3 | 81.6±0.4 | 67.4±0.3 | 97.4±0.0 | 98.1±0.1 | 67.8±0.7 | 89.9±0.3 | 50.8±0.2 | 43.4±0.6 | 88.2±0.1 |
| MW-MAE | 96.2±0.5 | 73.2±0.2 | 82.2±0.4 | 69.7±0.3 | 97.3±0.0 | 98.1±0.1 | 69.4±0.5 | 91.3±0.2 | 52.0±0.2 | 44.7±0.8 | 89.9±0.2 |

## H.1  ADDITIONAL WINDOW AND ATTENTION HEAD SIZE ABLATIONS

Although the proposed window size strategy in Sec 3.2 is quite straightforward and covers all potential windows for given number of input patches, we have conducted some additional ablations comparing

Table 12: Amount of pre-training dataset used v/s downstream performance.

| Model | BO | CD | ESC-50 | LC | Mri-S | Mri-T | NS-5h | SC-5h | F50K | VL | $s(m)$ |
|---|---|---|---|---|---|---|---|---|---|---|---|
| **10% of AS-5k** | | | | | | | | | | | |
| MAE | 93.6±0.7 | 51.3±0.2 | 49.5±0.3 | 48.4±0.4 | 97.1±0.1 | 96.4±0.1 | 61.1±0.7 | 70.4±0.9 | 29.7±0.2 | 17.3±0.5 | 63.3±0.2 |
| MW-MAE | 94.1±0.3 | 63.9±0.3 | 67.1±0.3 | 60.5±0.2 | 97.3±0.1 | 97.6±0.0 | 64.4±0.5 | 82.0±0.4 | 40.9±0.2 | 30.1±1.1 | 77.2±0.3 |
| **25% of AS-5k** | | | | | | | | | | | |
| MAE | 96.2±0.6 | 57.5±0.3 | 64.9±0.4 | 56.9±0.3 | 97.4±0.1 | 97.5±0.1 | 65.0±0.6 | 79.3±0.4 | 39.2±0.1 | 24.2±0.7 | 73.6±0.2 |
| MW-MAE | 96.1±0.5 | 68.0±0.2 | 75.5±0.4 | 67.2±0.3 | 97.3±0.1 | 98.0±0.1 | 65.9±0.4 | 86.5±0.2 | 46.4±0.1 | 35.7±0.6 | 83.8±0.2 |
| **50% of AS-5k** | | | | | | | | | | | |
| MAE | 97.2±0.3 | 65.5±0.3 | 74.1±0.3 | 64.3±0.3 | 97.5±0.1 | 98.1±0.1 | 67.0±0.6 | 85.3±0.6 | 45.1±0.1 | 32.4±0.8 | 81.9±0.2 |
| MW-MAE | 95.9±0.5 | 70.9±0.2 | 79.1±0.3 | 69.1±0.4 | 97.4±0.1 | 98.1±0.1 | 68.4±0.7 | 88.5±0.2 | 49.1±0.1 | 39.5±0.5 | 87.0±0.2 |
| **75% of AS-5k** | | | | | | | | | | | |
| MAE | 95.3±0.5 | 70.2±0.2 | 79.0±0.3 | 67.4±0.2 | 97.4±0.1 | 98.1±0.1 | 67.4±0.6 | 88.8±0.3 | 49.2±0.1 | 39.5±0.7 | 86.2±0.2 |
| MW-MAE | 96.0±0.5 | 72.6±0.3 | 80.5±0.4 | 69.5±0.3 | 97.4±0.1 | 97.9±0.1 | 68.3±0.4 | 89.9±0.2 | 50.5±0.1 | 41.7±0.8 | 88.4±0.2 |
| **100% of AS-5k** | | | | | | | | | | | |
| MAE | 96.2±0.3 | 72.2±0.2 | 80.9±0.4 | 67.3±0.3 | 97.4±0.1 | 98.3±0.1 | 68.3±0.4 | 89.4±0.3 | 50.4±0.1 | 43.1±0.9 | 88.1±0.2 |
| MW-MAE | 96.0±0.5 | 73.1±0.3 | 81.2±0.4 | 68.8±0.2 | 97.4±0.1 | 97.9±0.1 | 69.3±0.6 | 90.9±0.2 | 51.2±0.2 | 44.2±0.9 | 89.2±0.2 |

how performance treads with different window and attention head sizes in the decoder, as shown in Table 13. These experiments were done on the base configuration from the paper (ViT-B encoder, $n_p = 250$, decoder with 4 layers and 384 hidden dimensions).

Table 13: Ablation experiments on different window sizes and number of attention heads $h$. Irrespective of the window size/number of heads used, MW-MAE outperforms MAE in overall performance by a considerable margin.

| Model | BO | CD | ESC-50 | LC | Mri-S | Mri-T | NS-5h | SC-5h | F50K | VL | $s(m)$ |
|---|---|---|---|---|---|---|---|---|---|---|---|
| **h=4, Windows for MW-MAE: [5,25,125,250]** | | | | | | | | | | | |
| MAE | 96.1±0.4 | 71.9±0.1 | 80.2±0.5 | 67.6±0.3 | 97.5±0.1 | 98.0±0.1 | 66.7±0.4 | 89.7±0.2 | 49.8±0.1 | 42.8±0.9 | 87.6±0.2 |
| MW-MAE | 95.9±0.5 | 73.5±0.2 | 81.5±0.4 | 69.6±0.2 | 97.4±0.1 | 98.0±0.0 | 69.5±0.5 | 90.6±0.4 | 51.2±0.1 | 44.2±1.0 | 89.5±0.2 |
| **h=6, Windows for MW-MAE:[5,10,25,50,125,250]** | | | | | | | | | | | |
| MAE | 96.4±0.7 | 70.9±0.2 | 79.8±0.4 | 67.1±0.3 | 97.5±0.1 | 98.3±0.1 | 66.3±0.6 | 88.5±0.2 | 49.8±0.1 | 41.5±0.6 | 86.9±0.2 |
| MW-MAE | 96.2±0.5 | 73.3±0.4 | 81.3±0.3 | 68.4±0.2 | 97.2±0.1 | 97.9±0.1 | 68.8±0.6 | 90.3±0.3 | 51.1±0.1 | 44.7±0.9 | 89.1±0.2 |
| **h=8, Windows for MW-MAE: [2,5,10,25,50,125,250,250]** | | | | | | | | | | | |
| MAE | 96.2±0.3 | 72.2±0.2 | 80.9±0.4 | 67.3±0.3 | 97.4±0.1 | 98.3±0.1 | 68.3±0.4 | 89.4±0.3 | 50.4±0.1 | 43.1±0.9 | 88.1±0.2 |
| MW-MAE | 96.0±0.5 | 73.1±0.3 | 81.2±0.4 | 68.8±0.2 | 97.4±0.1 | 97.9±0.1 | 69.3±0.6 | 90.9±0.2 | 51.2±0.2 | 44.2±0.9 | 89.2±0.2 |
| **h=12, Windows for MW-MAE: [2,5,5,10,25,25,50,50,125,125,250,250]** | | | | | | | | | | | |
| MAE | 96.6±0.4 | 70.7±0.4 | 79.1±0.5 | 67.4±0.5 | 97.5±0.1 | 98.3±0.1 | 67.4±0.6 | 89.3±0.2 | 49.2±0.1 | 40.4±1.1 | 86.8±0.3 |
| MW-MAE | 96.3±0.6 | 73.6±0.3 | 81.0±0.4 | 68.9±0.2 | 97.5±0.0 | 98.0±0.1 | 70.3±0.9 | 91.1±0.2 | 50.9±0.1 | 44.2±0.9 | 89.5±0.3 |
| **h=16, Windows for MW-MAE: [2,2,5,5,10,10,25,25,50,50,125,125,250,250,250,250]** | | | | | | | | | | | |
| MAE | 96.5±0.4 | 70.7±0.1 | 77.1±0.2 | 67.8±0.3 | 97.5±0.1 | 98.4±0.0 | 68.2±0.6 | 88.5±0.3 | 48.5±0.2 | 38.7±0.9 | 86.4±0.2 |
| MW-MAE | 95.6±0.6 | 73.1±0.2 | 81.7±0.3 | 69.9±0.2 | 97.5±0.1 | 98.0±0.0 | 69.8±0.7 | 91.6±0.2 | 51.2±0.1 | 44.5±0.8 | 89.7±0.2 |

# I    LIMITATIONS

The direct limitations of our work are:

1. Pre-training data scale: As opposed to text corpus used in NLP Devlin et al. (2019) as well as speech representations Baevski et al. (2020); Hsu et al. (2021a), AudioSet is several order of magnitudes smaller. While MW-MAEs demonstrate good performance characteristics in low-data scenarios, analysis on larger scales of data would be beneficial. However, given the scope of this work (general-purpose audio representation learning), we firmly believe AudioSet is the best fit, as most of the larger audio corpora are speech only.

2. Computational demands: transformer based models are computationally expensive to train, and despite their favourable generalization characteristics, MW-MAEs are no different. MW-MAEs and as well as previous works Niizumi et al. (2022); Huang et al. (2022) have showed the efficacy of MAEs when pretrained with AudioSet, however, training on longer duration audio data is still a challenge.

3. Runtime Overhead: For our base configuration, the training time throughput of MW-MAE (4109 samples/ sec) is 9.14% slower than an MAE (4536 samples/sec) on the same hardware. This is primarily due to kernel call overhead of the MW-MHA module since we have to make individual calls for each attention head in an MW-MAE, whereas optimized MHA implementations do so in one kernel call. A better native implementation can alleviate this difference. However, since the decoder is discarded after pretraining, there is no difference in throughput when running downstream experiments/inference/finetuning since the encoders are identical.

