# OpenReview forum: "Masked Autoencoders with Multi-Window Local-Global Attention Are Better Audio Learners"
_ICLR.cc/2024/Conference — ICLR 2024 poster_

### Official Review · Reviewer_4eUM · 2023-10-30

**Soundness:** 2 fair
**Presentation:** 2 fair
**Contribution:** 3 good
**Rating:** 6
**Confidence:** 3

**Summary:**

This paper extends the masked auto-encoders by processing the input audio features with multiple windows of different sizes. The model is pre-trained  to reconstruct  the input features using MSE loss. Once the models are trained, the embeddings are passed through an MLP classifier to perform 10 different utterance-level speech tasks. As compared to the vanilla masked auto encoders, the proposed multi-window masked AE (MW-MAE) performs slightly better on those 10 tasks based on the overall score. If we look at each task separately, on some tasks the vanilla MAE performs better and in others the proposed MW-MAE performs better although the differences are usually small.

**Strengths:**

The main strengths are
1. Extensive experiments
2. Experimental details are provided in the Appendix (reproducibility)


- Originality and Significance:
This paper is an extension of the vanilla masked AE, and the proposal uses multi window approach to extract intermediate features at various scale (local to global). Multiscale approaches are not particularly new but this might be the first application of it to audio representation learning with masked AE.

- Quality:
Even though the proposed method is sound and extensive evaluations have been conducted for it, the performance gains due to using this model is limited as compared to the vanilla masked AE. Given other concerns about the run-time

- Clarity:
The paper is mostly easy to follow. However, it could be better to mention the tasks used in evaluation in the main text rather than leaving it for the Appendix.

**Weaknesses:**

1. Marginal gains with the proposed approach: Overall MW-MAE is slightly better than the original MAE. At individual task levels, sometimes MAE is better and sometimes MW-MAE is better by a small margin.

2. The local vs. global feature aspect of the model is not well-supported. From the multi-window approach, we can see that there are multiple scales but since there is no frame-level task in the evaluation setup, we cannot judge whether the local features/embeddings are also useful in downstream tasks. All the tasks evaluated here are based on global features at utterance level.

3. It could have been better to provide a quantitative run-time comparison of MAE and MW-MAE, rather than mentioning this shortcoming in the Appendix.

4. [Minor comment] It could have been better if the 10 tasks were mentioned in the main text before Table 1 rather than leaving the list to the Appendix.

**Questions:**

1) Which ten tasks have been used? (We cannot understand this without looking at the Appendix). Please explicitly mention these in the main text.

2) Evaluations are on utterance level tasks, so how can we make sure that the local information is also well-preserved at the frame level?

---

> ### Author Response · Authors · 2023-11-17
> **Author's Response to Reviewer 4eUM [Part 1/2]**
>
> We thank the reviewer for their comments. We have addressed your comments one by one.
>
> > Multiscale approaches are not particularly new but this might be the first application of it to audio representation learning with masked AE.
>
> Various approaches to local and global attention  have been applied to audio representation learning before (as several of our references indicate), but we have proposed a novel method with local-global windows within the same multihead module. Our approach is not merely novelty in application, it’s a new approach to local-global attention modelling.
>
> ## Weaknesses And Questions
>
> > 1. Marginal gains with the proposed approach .... MW-MAE is better by a small margin.
>
> Within the reported 95% confidence intervals, MW-MAEs consistently outperform MAEs in a host of conditions and scenarios.
>
> 1. The overall performance of MW-MAEs is considerably better than MAEs in a wide variety of conditions such as:
>     a. low data scenarios.
>     b. For larger encoder complexities: 92.6±0.2 vs 90.0±0.2 for ViT-L is a considerably large improvement, especially considering it’s an aggregation over 10 tasks.
> 2. At the task level,
>     a. MW-MAEs consistently outperform their MAE counterparts on majority of the tasks. For example, ViT-L based MW-MAE from table 2 achieves an absolute performance improvement of 2.3%, 2%, 1.2%, 2.2%, 1.8%, 1.7%, 5% on CD, ESC-50, LC, NS-5h, SC-5h, FSD50k and VL tasks, respectively. That is a considerable improvement across the board on 7/10 tasks.
>     b. The only tasks it does not improve on, are BO, Mri-S and Mri-T, where MW-MAEs are within the reported 95% confidence intervals, not worse. This observation is mostly consistent across all our experiments. We suspect this is due to performance saturation on these tasks, as also suggested by the HEAR leaderboard [a].
>
> > 2. The local vs. global feature aspect of the model is not well-supported...global features at utterance level.
>
> Before continuing, we would like to add that just because a task is evaluated at utterance level, it doesn’t mean that local information was not leveraged at all. For example, FSD50K is a multilabel audio tagging task, with multiple audio events occurring throughout the audio clip. While prediction is done at the utterance level, rich local feature information is crucial for detecting all the audio events present in the audio clip. This local information is consolidated over the layers and assimilated into a feature representation. Also, we have shown extensive, quantitative exploratory analysis in Section 5.1 that verifies that the proposed MW-MAE captures a better mix of local and global attention.
>
> Having said that, we appreciate the reviewer's suggestion. We have conducted additional experiments comparing MAE and MW-MAE base configurations on MAESTRO-5h, a frame-level music transcription task, which is a part of the HEAR benchmark. We chose MAESTRO-5h over DCASE 2016 Task 2 because our test suite includes ESC-50 and FSD50K, which already span office sound environments. It is evaluated in a 5-fold cross-validation setting, with the Onset FMS evaluation metric. We conducted the experiments 5 times and reported 95% confidence intervals over the cross-validation scores of individual runs.
>
> | Model | Onset FMS |
> | --- | --- |
> |  |  |
> | MAE-Base | 46.47±0.05 |
> | MW-MAE Base | 47.31±0.07 |
>
> MW-MAE does improve performance on MAESTRO-5h. While this might not look like much, at the time of writing this comment, **MW-MAE outperforms every method on the HEAR leaderboard [a] for MAESTRO-5h, including large multi and cross-modal ensemble methods**, so this is excellent performance for a single method. For reference, a pre-trained Wav2Vec2-Base only achieves an Onset FMS of 3.29 [a].
>
> We didn’t compare all methods on MAESTRO to begin with, because it poses serious computation (and space) complexity: it takes longer to evaluate downstream performance on just MAESTRO than it takes to evaluate all the other downstream tasks combined. For the number of experiments we did in this paper, it was highly unfeasible, more so given that our benchmark suite includes tasks like FSD50K, where local information is known to be of importance [b].
>
> Also, in response to reviewer fDLB, we have also posted proof of concept results for Supervised ASR, which might be of relevance in regard to this question.
>
> ---
>
> Continued..

---

> ### Author Response · Authors · 2023-11-17
> **Author's Response to Reviewer 4eUM [Part 2/2]**
>
> Continued..
>
> ---
>
> > 3. It could have been better to provide a quantitative run-time comparison...
>
> For our base configuration, the _training time_ throughput of MW-MAE (4109 samples/ sec) is 9.14 % slower than an MAE (4536 samples/sec) on the same hardware. This is primarily due to kernel call overhead of the MW-MHA module since we have to make individual calls for each attention head in an MW-MAE, whereas optimized MHA implementations do so in one kernel call.  A better native implementation can alleviate this difference, and we are working on that.
>
> However, it is worth noting that since the decoder is discarded after pretraining, _there is no difference in throughput when running downstream experiments/inference/finetuning_ since the encoders are identical.
>
> > 4. [Minor comment] It could have been better if the 10 tasks were mentioned in the main text before Table 1 rather than leaving the list to the Appendix.
>
> That is an oversight on our behalf. We will mention the 10 tasks in the main text, however, the table will remain in the Appendix due to space limitations.
>
> ---
>
> ## Conclusion
>
> We thank the reviewer for the valuable feedback and for engaging in a discussion with us! We hope we have addressed your concerns, and we would love to hear back!
>
> ## References
> ---
>
> [a] https://hearbenchmark.com/hear-leaderboard.html
> [b] Schmid et al, 2023, “Dynamic Convolutional Neural Networks as Efficient Pre-trained Audio Models”

---

> > ### Author Response · Authors · 2023-11-20
> > **A gentle nudge to Reviewer 4eUM**
> >
> > Dear Reviewer 4eUM,
> >
> > We have responded to your review and have incorporated your suggestions as well as conducted additional experiments to address your concerns. The discussion phase is ending soon, and we would appreciate if you could go through our response so that we can respond to any further questions you might have.
> >
> > Best,
> > Authors

---

> > > ### Comment · Reviewer_4eUM · 2023-11-22
> > > **Update to the previous score**
> > >
> > > I would like to thank the authors for adding clarification and for running additional experiments. The new results support the claim that the proposed method consistently outperforms the baseline approach.
> > >
> > > As for Q2, I agree that an utterance level task can still capture the local information but the presentation in the paper did not include an explicit evidence in its initial version. Once again, thanks for running the additional experiments.
> > >
> > > As for Q3, I was concerned about the inference run-time rather than training throughput, however, I can see that the authors' comment about kernel level implementations for speed would be applicable there also.
> > >
> > > As a result, I am increasing my score to marginally above the threshold.

---

> > > > ### Author Response · Authors · 2023-11-22
> > > >
> > > > Thank you so much for the vote of confidence!
> > > >
> > > > > As for Q2, I agree that an utterance level task can still capture the local information but the presentation in the paper did not include an explicit evidence in its initial version. Once again, thanks for running the additional experiments.
> > > >
> > > > We wholeheartedly saw your concern, and were happy to comply!
> > > >
> > > > Overall, we thank the reviewer for taking the time to review our paper and provide essential feedback! We believe it has immensely improved the paper. And finally, thanks for investing time and energy to go through our response and the additional experiments!

---

### Official Review · Reviewer_fDLB · 2023-11-01

**Soundness:** 2 fair
**Presentation:** 3 good
**Contribution:** 1 poor
**Rating:** 6
**Confidence:** 2

**Summary:**

This paper introduces a Multi-Window Masked Autoencoder (MW-MAE) tailored to effectively capture both local and global time-frequency details in audio. The approach applies self-attention over non-overlapping windows of different sizes for each head within the multi-head attention of transformer blocks. The authors show that the proposed method outperforms the standard Masked Autoencoders (MAEs) in ten subsequent audio downstream tasks and exhibit improved scaling traits with an increasing number of patches or model parameters.

**Strengths:**

* The paper is well-structured and easy to follow.
* The authors have made their implementation publicly available, which not only validates their results but also contributes to the research field.
* The proposed approach to enhancing multi-head attention is straightforward, yet it demonstrates a performance boost.
* The authors provide both quantitative and qualitative evidence to show that their proposed method effectively models both global and local information of data.

**Weaknesses:**

Limited applicability and contribution. The proposed method involves applying self-attention over non-overlapping windows of varying sizes for each head in the multi-head attention. However, experimentally, the method is only applied within the confines of a self-supervised approach in the audio domain, using the masked autoencoder structure, and specifically for the decoder. Compared to the previous work, Audio-MAE, which addressed challenges in initially applying MAE to the audio domain, this study merely demonstrates the effects of improving a specific part of the structure.

**Questions:**

* I have concerns regarding the limited applicability and contribution of this research, as mentioned in the above weaknesses.
* Typo: (6p, Sec. 4.3.) ... with the largest "MW-MAE-L-16x4-8l" -> ...... with the largest "MW-MAE-L-4x16-8l"

---

> ### Author Response · Authors · 2023-11-17
> **Author's Response to Reviewer fDLB**
>
> We thank the reviewer for their comments!
>
> > The paper is well-structured and easy to follow.
>
> We are glad that the reviewer found the paper easy to read! We gave it our best!
>
> > The proposed approach to enhancing multi-head attention is straightforward, yet it demonstrates a performance boost.
>
> Thanks for the words of encouragement! It elates us that the reviewer also sees the strength in our simple yet effective approach!
> Now, allow us to address your concerns!
>
> ---
>
> ## Concerns regarding applicability and contribution
>
> > The proposed method involves applying self-attention over .... using the masked autoencoder structure, and specifically for the decoder.
>
> ### [Proof of concept 1: Auxiliary Modality] MW-MAEs on ImageNet
>
> We would like to bring to your attention our initial experimental results using Masked Autoencoders for computer vision on the ImageNet dataset (as presented in Appendix D). We pretrained MAE and MW-MAEs with ViT-Medium encoder and a 4-layer decoder with 8 attention heads and 512 feature dimensions on ImageNet for 100 epochs, followed by evaluating linear probe performance (training for 50 epochs). MW-MAE outperforms MAE by almost 1% in validation accuracy.
>
> | Model | Params | Validation Acc |
> | --- | --- | --- |
> | MAE- Medium Encoder | 38 M | 29.0±0.0 |
> | MW-MAE Medium Encoder | 38 M | 29.8±0.0 |
>
> ### [Proof of concept 2: Auxiliary Task] MW-MHA for Supervised ASR
>
> We have conducted some preliminary experiments to investigate the applicability of the proposed MW-MHA module beyond MAEs. We evaluated Supervised Transformer based ASR models, trained using CTC objective function. We trained on LibriSpeech 100h subset and reported Word Error Rates (WER) on the test sets, using validated recipes from the SpeechBrain toolkit. We will add these results to the appendix.
>
> - The network architecture used consists of a 2-layer Convolutional frontend, followed by a 12-layer encoder and 4-layer decoder, each with 4 attention heads. We trained on mel-filterbanks with 80 mel bins.
> - For the MW-MHA Transformer, we only replaced the MHA blocks in the encoder with MW-MHA blocks, and used attention window sizes of 1/8, 1/4, 1/2 and global attention corresponding to the number of timeframes obtained from the largest audio segment in the corpus.
> - MW-MHA transformer outperforms baseline transformer by a considerable margin on both test sets.
>
> | Model | Params | Test Clean WER | Test Other WER |
> | --- | --- | --- | --- |
> | Transformer | 8.7 M | 7.56 | 17.90 |
> | MW-MHA Transformer | 8.7 M | 6.73 | 17.20 |
>
> While the scope of our investigation in this paper is confined to self-supervised learning, audio and masked autoencoders, these two auxiliary experiments clearly indicate that the proposed approach has the potential for wide applicability!
>
> ---
>
> > Compared to the previous work, Audio-MAE, which addressed challenges in initially applying MAE to the audio domain, this study merely demonstrates the effects of improving a specific part of the structure.
>
> Audio-MAE addressed how global attention is suboptimal for spectrograms if the time-frequency information of interest is predominantly local, and to this end, they proposed to incorporate SWIN [a] based attention in some transformer blocks of _only the decoder_ of their MAE (similar to us!). We are addressing the same challenges as AudioMAE did!
>
> Furthermore, compared to AudioMAE:
>
> 1. We perform more exhaustive evaluation of patch sizes, which is a key hyper-parameter that differs between MAEs for Vision v/s audio, as demonstrated by [b]. [b] precedes the AudioMAE paper and, chronologically, were the first to address the nuances of applying masked autoencoders to audio.
> 2. We evaluate on a larger, more varied selection of tasks, addressing a wider variety of challenging applications in the audio domain.
> 3.  We also conducted direct in-depth exploratory analysis of the local-global effects in the audio domain, measuring quantifiable variances in attention distances and entropies, which was missing from AudioMAE.
>
> Thus, we believe our contributions are just as significant, as we go above and beyond in several facets of investigating the challenges of applying MAE to the audio domain.
>
> ---
>
> ## Questions
>
> > I have concerns regarding the limited applicability and contribution of this research, as mentioned in the above weaknesses.
>
> Addressed above.
>
> > Typo: (6p, Sec. 4.3.) ...
>
> Thank you for bringing this to our attention! We will fix this.
>
> ---
>
> ## Conclusion
>
> Again, we thank the reviewer for their input and for engaging in a discourse! We hope we were able to address your concerns and look forward to answering any more questions!
>
> ---
>
> ## References
>
> [a] Liu et al. 2021, “Swin transformer: Hierarchical vision transformer using shifted windows.”
> [b] Niizumi et al. 2022, "Masked Spectrogram Modeling using Masked Autoencoders for Learning General-purpose Audio Representation”.

---

> ### Author Response · Authors · 2023-11-20
> **A gentle nudge to reviewer fDLB**
>
> Dear Reviewer fDLB,
>
> We have responded to your review, incorporated your suggestions, as well as added additional experiments w.r.t. your concerns. Given that the discussion period will end soon, we would appreciate it if you could go through our response, and maybe we can discuss any further questions you might have!
>
> Best,
> Authors

---

> > ### Comment · Reviewer_fDLB · 2023-11-22
> > **Comments on Author's Response**
> >
> > I am grateful to the authors for addressing my concerns to a certain extent. I acknowledge that this research provides sufficiently comprehensive experiments and analysis. The results of the auxiliary experiments they provided demonstrate that the proposed method is effective not only in self-supervised learning for audio data but also in ASR. Consequently, I would like to revise my rating to a 6. However, I still retain some reservations, particularly since many challenges addressed in this work were previously tackled in AudioMAE and its concurrent works. Additionally, as mentioned in the "Author's Response to Reviewer 4eUM," the increase in training time due to the introduced architecture somewhat negates the efficacy of the proposed method.

---

> ### Author Response · Authors · 2023-11-22
>
> > The results of the auxiliary experiments they provided demonstrate that the proposed method is effective not only in self-supervised learning for audio data but also in ASR
>
> Thank you so much! It feels great that you are also enthusiastic about the applicability of our approach!
>
> >  Additionally, as mentioned in the "Author's Response to Reviewer 4eUM," the increase in training time due to the introduced architecture somewhat negates the efficacy of the proposed method.
>
> The performance reduction in training time is solely due to implementation: Multi Head Attention has gone through several rounds of optimization of the kernel implementation in low-level code (CUDA, ROCm etc) and calls all attention heads in a single kernel call, which significantly optimizes issues such as memory transfer and kernel overheads. Our implementation as of now, as we pointed out to reviewer 4eUM, calls underlying attention heads individually which leads to overhead. We have complete confidence that a better implementation of native windowed attention can iron out these issues!
>
> ## Conclusion
>
> Overall, we would like to thank the reviewer for taking the time to review our paper, and providing feedback which has proved of immense benefit, and finally, for investing the time to go through our rebuttal and additional experiments.

---

### Official Review · Reviewer_a4K8 · 2023-11-09

**Soundness:** 2 fair
**Presentation:** 3 good
**Contribution:** 2 fair
**Rating:** 6
**Confidence:** 3

**Summary:**

This paper proposed a novel Multi-Window Multi-Head Attention module and replaced the Multi-Head Attention module by it within Masked AutoEncoder. The authors proved this new model Multi-Window Mased AutoEncoder (MW-MAE) as a better audio learner than MAE by empirical results on 10 downstream audio tasks. The authors attributed the improvement on that MW-MAE encoder learn heads with broader local and global attention, and then utilized attention entropies and distances analysis to support this argument. The authors utilized CCA to demonstrate that Multi-Window Multi-Head Attention in the decoder can independently capture local and global information.

**Strengths:**

1. The paper proposed a novel module MW-MHA that combines local and global attentions on attention heads level. It also utilized all non-unary factors to decide the number of heads, which is interesting.
2. The paper is well structured and includes detailed ablation study results and inplementation settings.
3. The paper adopted the proper analysis to support MW-MAE's strengths and adopted proper plot to elaborate analysis results.

**Weaknesses:**

1. Typos such as "we use fixed sinusuidal positional embeddings"
2. lack of parameter explanations in 3.1
3. Another work [1] proposed very similar ideas that introducing local attentions in the decoder can help improve the MAE learner. They tried local attentions only and hybrid attentions in which they applied global attentions on top layers. Although This paper compared MW-MAE with audioMAE and MW-MAE outperformed audioMAE by a lot, it is still unclear to me why MW-MAE is much better with such a similar design. I attached my questions in below.
4. Introduction section and Related Works section overlaped by a lot.
5. There should be another line "enc only" in table 2.


[1] Huang et al, “Masked Autoencoders that Listen,” NeurIPS 2022

**Questions:**

1. I'm wondering whether the results of audioMAE and MW-MAE are comparable, what is the patch size and spectrogram size of audioMAE you are using? Have you tried re-training audioMAE with your current settings?
2. According to paper [1], their number of audioMAE(local) on ESC-50 was 94.1 and your number of audioMAE on ESC-50 is 60.6, what is the difference?

---

> ### Author Response · Authors · 2023-11-18
> **Author's Response to Reviewer a4K8**
>
> We thank the reviewer for their comments. We are happy that the reviewer finds our analysis in Sec 5 appropriate! Let us address your concerns.
>
> ## Weaknesses
>
> > 1. Typos such as "we use fixed sinusuidal positional embeddings"
>
> Thanks for bringing this to our attention! We’ll fix this and comb through the content for typos again.
>
> > 2. lack of parameter explanation in 3.1
>
> $Q, K, V \in \mathbb{R}^{n \times {d_m}}$ represent query, key and value input matrices, and $W_i^Q, W_i^K, W_i^V \in \mathbb{R}^{d_m \times d_k}$ are their corresponding learnable projection matrices. $d_m$ is the model's feature dimension, and $d_k = \frac{d_m}{h}$.
>
> We’ll add this to the main text in Sec 3.1
>
> > 3. Another work [1] proposed very similar ideas that introducing local attentions in the decoder can help improve the MAE learner. They tried local attentions only and hybrid attentions in which they applied global attentions on top layers. Although This paper compared MW-MAE with audioMAE and MW-MAE outperformed audioMAE by a lot, it is still unclear to me why MW-MAE is much better with such a similar design. I attached my questions in below.
> >
>
> [a], which precedes AudioMAE, presented extensive experiments for training masked autoencoders on Audioset. Early on, we observed that their approach performed better than AudioMAE, and we thus build upon [a] to build a stronger baseline than AudioMAE. Key differences include:
>
> 1. For the base configuration, AudioMAE has a patch size of 16x16. Our MAE baseline has a patch size of 16x4, which gives better temporal resolution.
> 2. AudioMAE is trained on full 10-sec audio clips, whereas our models are trained on random 2-second crops, based on the published results of [a] where 2-sec crops offer a great performance/training time tradeoff.
>
> We have already discussed these design decisions in section 4.2.
>
> > Have you tried re-training audioMAE with your current settings?
> >
>
> This stronger MAE baseline is used for fair, direct comparison with the proposed MW-MAE approach. Since we already had a stronger baseline, for consistency and being truthful to the source, we decided to conduct downstream experiments for AudioMAE using the official pre-trained model and code.
>
> > 4. Introduction section and Related Works section overlapped by a lot.
>
> Introduction lays the ground work of the relevant literature & the Related works section expands on it with commentary on specific approaches. We will clear up repetitive references & explanations.
>
> > 5. There should be another line "enc only" in table 2.
>
> Thank you for the suggestion. We have now also included this extra experiment on MW-MAE Base (encoder only) setting. We didn’t believe originally that it was necessary, given that enc+dec doesn’t provide substantial improvements for pre-training. Following will be the updated Table 2:
>
> | Model | Downstream $s(m)$ | Linear Probe (mAP)  | Fine-tuning (mAP) |
> | --- | --- | --- | --- |
> |  |  |  |  |
> | MAE Base | 88.1±0.2 | 23.1±0.0 | 26.1±0.4 |
> | MW-MAE Base (decoder only) | 89.2±0.2 | 24.2±0.1 | 26.1±0.7 |
> | MW-MAE Base (encoder only) | 89.1±0.3 | 23.6±0.0 | 26.1±0.3 |
> | MW-MAE Base (enc+dec) | 89.1±0.3 | 24.0±0.1 | 26.2±0.0 |
>
> While there is no difference in performance when fine-tuning, there is a considerable decline in performance in the encoder only setting for linear probe performance. The overall conclusion from the paper, regarding the addition of MW-MHA modules, doesn’t change.
>
> ---
>
> ## Questions
>
> > 1. I'm wondering whether the results of audioMAE and MW-MAE are comparable, what is the patch size and spectrogram size of audioMAE you are using? Have you tried re-training audioMAE with your current settings?
> >
>
> we have answered this in the section above.
>
> > 2. According to paper [1], their number of audioMAE(local) on ESC-50 was 94.1 and your number of audioMAE on ESC-50 is 60.6, what is the difference?
> >
>
> ESC-50 number reported in AudioMAE is a result of supervised fine-tuning of the entire encoder. This is different from our setting, where we follow the HEAR [b] protocol and conduct downstream experiments by training an MLP on top of fixed feature vectors, and do not fine-tune the entire encoder. This is inline with recent approaches for evaluating self-supervised audio representations [b, c].
>
> ---
>
> ## Conclusion
>
> We thank the reviewer for engaging in the discussion and hope we were able to address your concerns. We look forward to answering any further questions the reviewer might have!
>
> ---
>
> ## References
>
> [a] Niizumi et al., 2022, “Masked spectrogram modeling using masked autoencoders for learning general-purpose audio representation”.
> [b] Turian et al., 2022, “HEAR: Holistic Evaluation of Audio Representations”
> [c] Yang et al., 2021, “SUPERB: Speech Processing Universal PERformance Benchmark”

---

> > ### Comment · Reviewer_a4K8 · 2023-11-20
> > **Comments on Author's Response**
> >
> > Thank you so much for addressing my concerns, I have few more questions about your comments.
> > >*"[a], which precedes AudioMAE, presented extensive experiments for training masked autoencoders on Audioset. Early on, we observed that their approach performed better than AudioMAE, and we thus build upon [a] to build a stronger baseline than AudioMAE"*
> >
> > Is there any reference that proves [a] is a better baseline than AudioMAE under the same settings? Intuitively, AudioMAE could be a necessary baseline since your conclusion is similar with AudioMAE's, which is adding local/hybrid attention in decoder can help the encoder generate better representations. Based on the similarities, I'm wondering how much that MW-MAE outperforms AudioMAE under the same settings and what makes MW-MAE outperforms AudioMAE. If the gap between MW-MAE and AudioMAE only comes from patch size and spectrogram size difference, the novelty should be considered marginal.

---

> ### Author Response · Authors · 2023-11-20
>
> | Is there any reference that proves [a] is a better baseline than AudioMAE under the same settings?
>
> In paper [a], Table 5, authors conducted downstream experiments on several HEAR datasets with the same patch size as AudioMAE (16x16) and 2-second audio input: MSM-MAE-208.
>
> Early on in our study before we committed to a reference implementation to build upon, we basically compared these numbers with the results we obtained using official AudioMAE pre-trained weights and found that AudioMAE downstream results were worse than MSM-MAE-208. For instance, AudioMAE gave us 68.2±0.2 and 37.9±0.1 for CD and FSD50K. For the same tasks, MSM-MAE numbers from [a] were 70.4 and 50.8, respectively. Considering the efforts that go into reproducing a paper, the decision to build upon the stronger baseline seemed justified to us. And thus, the performance gap cannot be chalked merely to difference in patch and spectrogram size.
>
> | I'm wondering how much that MW-MAE outperforms AudioMAE under the same settings
>
> We understand your concern and point of view. We have been running experiments comparing AudioMAE (essentially SWIN Transformer in the decoder) with our proposed approach under identical settings: smaller patch size than original AudioMAE, smaller decoder than original AudioMAE (4 layers instead of 16), 2-second audio crops vs 10. We'll get back to you once we have the results.

---

> ### Author Response · Authors · 2023-11-21
> **Comments on Reviewer's response: additional experiments on AudioMAE in identical setting.**
>
> Dear reviewer a4K8,
>
> As we mentioned in our previous comment, we have conducted additional experiment on AudioMAE (essentially SWIN blocks in the decoder) in identical setting as the proposed MW-MAE for a direct comparison. We conduct experiments for the base configuration: ViT-Base encoder, 4x16 patch size, decoder with 4 layers and 384 hidden dimensions.
>
> Following are the results on the different tasks. We have added results of comparable MW-MAE method from the main text for easy viewing.
>
> | model              | BO   | CD   | ESC-50   | LC   | Mri-S   | Mri-T   | NS-5h   | SC-5h   | FSD50k   | VL   | $s(m)$   |
> |:-------------------|:----------------|:----------|:---------|:-------------|:--------------|:--------------|:------------------|:--------------------|:---------|:------------|:-------------------|
> | AudioMAE-B-4x16-4l | 96.0±0.5        | 72.4±0.3  | 72.0±0.5 | 66.9±0.4     | 97.2±0.0      | 98.2±0.1      | 69.8±0.8          | 89.8±0.3            | 49.0±0.1 | 38.3±0.8    | 86.1±0.3           |
> | MAE-B-4x16-4l      | 96.2±0.3        | 72.2±0.2  | 80.9±0.4 | 67.3±0.3     | 97.4±0.1      | 98.3±0.1      | 68.3±0.4          | 89.4±0.3            | 50.4±0.1 | 43.1±0.9    | 88.1±0.2           |
> | MW-MAE-B-4x16-4l   | 96.0±0.5        | 73.1±0.3  | 81.2±0.4 | 68.8±0.2     | 97.4±0.1      | 97.9±0.1      | 69.3±0.6          | 90.9±0.2            | 51.2±0.2 | 44.2±0.9    | 89.2±0.2           |
>
> Compared to the official AudioMAE pretrained model [1], retraining on 2-sec inputs and 4x16 patch size improves AudioMAE performance drastically. However, MW-MAEs ($s(m)=89.2±0.2)$ still demonstrate a considerable absolute improvement of 3.1% in overall score over the AudioMAE model ($s(m)=86.1±0.3$).
>
> Given that we have now compared the two methods in identical settings, we can conclude that this performance gap between AudioMAE and MW-MAE is due to better local-global attention modelling capabilities of the proposed approach.
>
> We thank the reviewer for the suggestion, as this further demonstrates the strength of our approach. We hope this addresses your concerns and we eagerly wait to hear back from you.
>
> *EDIT: Added MAE results as well.
>
> ---
>
> ## References
> [1] Huang et al., 2022, "Masked Autoencoders that Listen"

---

> > ### Comment · Reviewer_a4K8 · 2023-11-21
> > **Comments on additional AudioMAE experiments**
> >
> > Thank you for response, does this result indicate that MAE with SWIN blocks in decoder is worse than Vanilla MAE on these 10 tasks?

---

> ### Author Response · Authors · 2023-11-22
> **Response to reviewer's Comments on additional AudioMAE experiments**
>
> We have added the MAE results in the table in our previous comments. You can see that MAE with Swin blocks performs worse than our MAE baseline on some tasks (4/10) and at par on others. MAE with Swin blocks does have considerably worse overall performance than our MAE baseline, although not as much of a disparity compared to official pre-trained AudioMAE model.

---

> > ### Comment · Reviewer_a4K8 · 2023-11-22
> > **Modification on rating**
> >
> > I would like to modify the rating to 6 regarding the novelty of MWMHA if your paper is the first one to propose this module. On the other hand, I think this specific application of MWMHA on MAE could be much restricted.

---

> ### Author Response · Authors · 2023-11-22
>
> We conducted an extensive literature review, and to the best of our knowledge, we are the first to propose a multi-windowed multihead attention module as presented in the paper.
>
> > On the other hand, I think this specific application of MWMHA on MAE could be much restricted
>
> We would like to bring to the reviewer's attention, in case they haven't seen our response to Reviewer fDLB, our preliminary results investigating MW-MHA for Supervised ASR. For more details, you can view our response to Reviewer fDLB, but the gist of it is that while true that it is just preliminary experiments, we observed considerable improvement in WER performance on LibriSpeech, which highlights potential applicability of MW-MHA beyond MAEs.
>
> | Model | Params | Test Clean WER | Test Other WER |
> | --- | --- | --- | --- |
> | Transformer | 8.7 M | 7.56 | 17.90 |
> | MW-MHA Transformer | 8.7 M | 6.73 | 17.20 |
>
> ## Conclusion
>
> We thank the reviewer for their suggestions, and most of all, for continuously engaging in discussions with us! We had a lot of work cut out for us (justifiably) for the rebuttal based on reviewer feedback, so hearing back from reviewers really bolsters our confidence! Thanks for taking the time to review our paper.

---

> ### Comment · Reviewer_a4K8 · 2023-11-22
> **Comments on authors' response**
>
> Thank you for these additional experiments and figures, now I understand that MWMHA has some potential on other tasks. However, the main topic of this paper is the application of MWMHA in MAE on the audio representation task, and the improvements from this specific application of MWMHA is not as much as those from other tasks.  I'd like to clarify my statement, I think the effectiveness of this specific application of MWMHA in MAE is restricted.

---

### Official Review · Reviewer_M3PT · 2023-11-09

**Soundness:** 2 fair
**Presentation:** 2 fair
**Contribution:** 2 fair
**Rating:** 3
**Confidence:** 3

**Summary:**

The authors propose the multi-window multi-head attention module to model the local-global interactions in deocder transformer for masked autoencoders.

**Strengths:**

The multi-window local-global attention is novel for masked autoencoder learning.

**Weaknesses:**

1.The conceptualization of this article is relatively straightforward. Building upon the previous global-local attention mechanism, previous studies did not incorporate a hierarchical design for the local window perspective. Hence, the author implemented a mechanism with varying window sizes for each attention head within the multi-head attention mechanism. This initial approach is commendable. I am intrigued to know if the author is the originator of this multi-window local-global attention concept for the first time. No pertinent references were found at the end of the introduction, maybe leaving open the possibility of existing similar ideas in the fields of computer vision or natural language processing.

2.In terms of the experimental aspect, "the default configuration employed by the authors yields np = 250, resulting in window sizes of [2, 5, 10, 25, 50, 125, 250, 250] for each MW-MHA module in all decoder blocks, encompassing a total of eight attention heads." Although the author clarifies that this hyperparameter design covers several possible local context levels and adopts a simplistic set of designs, further ablation experiments pertaining to this aspect would be valuable.

3.Corresponding to the points highlighted by the author in Appendix G, the experiments conducted solely focused on the (AS-5k) dataset. However, as a self-supervised model, it is expected that the author would train on larger datasets, given the significance of datasets such as librilight, which have already amassed 60k hours of data. Furthermore, exploring the model's performance on larger datasets would provide insights into its upper limits.

4.A meticulous examination of the author's ablation experiments on the parameters of MAE in the appendix reveals that, under different parameter configurations and tasks, MW-MAE does not exhibit significant improvements compared to MAE. For instance, [BO, Patch Size=(4×8), n=500, h=12, MAE 96.7±0.2, MW-MAE 95.6±0.7], [Mris-s all], [Mri-T all]. Similar observations can be made in the main experiments presented in Table 1. In certain tasks, a comprehensive enhancement is not evident. Consequently, I eagerly anticipate the author's elucidation regarding this aspect.

**Questions:**

The overall engineering effort invested in this paper appears to be relatively limited, and there are certain limitations in the analysis of the multi-window multi-head attention. Could more diverse designs and comparisons be explored, specifically regarding local attention windows? For instance, is it possible to make these window sizes trainable? Alternatively, more engaging designs and comparisons could be conducted by examining the relationship between the number of attention heads, network hierarchies, and window sizes.

---

> ### Author Response · Authors · 2023-11-16
> **Author's Response to Reviewer M3PT (1/2)**
>
> We thank the reviewer for their comments. Allow us to address your concerns one by one
>
> ## Weakness 1
>
> > Hence, the author implemented a mechanism with varying window sizes for each attention head within the multi-head attention mechanism. This initial approach is commendable.
> >
>
> We are excited that the reviewer finds our approach commendable! Thanks for the words of encouragement.
>
> > I am intrigued to know if the author is the originator of this multi-window local-global attention concept for the first time
> >
>
> We conducted an extensive literature review, and to the best of our knowledge, we are the first to propose a multi-windowed multihead attention module as presented in the paper.
>
> > No pertinent references were found .... in the fields of computer vision or natural language processing.
> >
>
> We have included key pertinent references for various related approaches in the introduction (such as sliding window attention-based methods [a-c] as well as multiscale attention-based methods [d-g]). We also elucidate how the proposed approach differs from existing approaches in Section 3.1. These references span domains beyond audio representation learning.
>
> ## Weakness 2
>
> > In terms of the experimental aspect ..... further ablation experiments pertaining to this aspect would be valuable.
> >
>
> Since this weakness overlaps with the question posed by the reviewer, we have addressed both these issues here. We have already conducted extensive experiments w.r.t. network hierarchy (as per Sec 4.4). Thus, we focus on further ablations exploring the downstream performance impact of different window sizes and the number of attention heads, as shown in Table below. All evaluated methods have an identical encoder (ViT-Base, 4x16 patch size), number of patches $n_p = 250$, the same number of decoder layers ($l=4$) with the same hidden dimension ($d=384$). We will add detailed results to the appendix (we were having trouble squeezing in the full markdown table in a textbox).
>
>
> | model | n_heads | Windows | score |
> | --- | --- | --- | --- |
> | MAE Base | 4 | N/A | 87.6±0.2 |
> | MW-MAE Base | 4 | [5,25,125,250] | **89.5±0.2** |
> |     |    |     |      |
> | MAE Base | 6 | N/A | 86.9±0.2 |
> | MW-MAE Base | 6 | [5,10,25,50,125,250] | **89.1±0.2** |
> |     |    |     |      |
> | MAE Base | 8 | N/A | 88.1±0.2 |
> | MW-MAE Base | 8 | [2,5,10,25,50,125,250,250] | **89.2±0.2** |
> |     |    |     |      |
> | MAE Base | 12 | N/A | 86.8±0.3 |
> | MW-MAE Base | 12 | [2,5,5,10,25,25,50,50,125,125,250,250] | **89.5±0.3** |
> |     |    |     |      |
> | MAE Base | 16 | N/A | 86.4±0.2 |
> | MW-MAE Base | 16 | [2,2,5,5,10,10,25,25,50,50,125,125,250,250,250,250] | **89.7±0.2** |
>
> As evident, irrespective of the window size used, MW-MAE outperforms MAE in overall performance by a considerable margin.
>
> ## Weakness 3
>
> > Corresponding to the points highlighted .... exploring the model's performance on larger datasets would provide insights into its upper limits
> >
>
> While we empathize with the reviewer's point of view, there are several reasons to stick to AudioSet. Most larger audio datasets, such as those suggested by the reviewer (LibriLight) and others (e.g. VoxPopuli), comprise only speech data, which does not align with the scope of this paper (general-purpose audio representation learning). For learning audio representations that go beyond speech, AudioSet has been a defacto standard in the community, as it consists of in-the-wild audio spanning an ontology of 527 sounds such as human speech, vehicles, everyday objects, and environmental and animal sounds, to name a few.  AudioSet has been used by several key approaches for unsupervised representation learning, such as COLA [h], BYOL-A [i], SSaST and AudioMAE. For consistent comparison in line with existing literature as well as alignment with the scope of our work, picking AudioSet was the ideal choice.
> Our motivation for evaluating MAEs v/s MW-MAEs when using different amounts of pre-training data (Section 4.4) was to simulate how performance treads with dataset scale, where MW-MAEs outshine MAEs consistently.
>
> ---
>
> Continued..

---

> ### Author Response · Authors · 2023-11-17
> **Author's Response to Reviewer M3PT (2/2)**
>
> Continued ...
>
> ---
>
> ## Weakness 4
>
> > A meticulous examination of the author's ablation ...... author's elucidation regarding this aspect.
> >
>
> Even in the worst cases, MW-MAEs perform on par (based on the presented 95% confidence intervals) with MAEs, including the cases highlighted by the reviewer. This phenomenon is observed only for BO, Mri-S and Mri-T tasks, as also discovered by the reviewer. Based on our observations for larger model configurations, we believe performance on these datasets to be saturated around this upper range (for e.g. no model breaks 98% mark on Mri-S, not even the very large ViT-L and ViT-H configurations). This is also supported by observations from the HEAR leaderboard [j], where even multi-modal ensembles of several large models do not go beyond this performance.
>
> Throughout the paper, we have shown exhaustive evidence that shows that MW-MAEs consistently outperform MAEs by a considerable margin on other, more difficult tasks (such as FSD50k: 53.5±0.1 vs 51.8±0.1 for ViT-L encoder), as well as in overall performance (92.6±0.2 vs 90.0±0.2 for ViT-L). Within 95% confidence intervals, MW-MAEs outperform MAEs consistently on 7/10 tasks while being on par on the other 3 tasks. For example, ViT-L-based MW-MAE from table 2 achieves an absolute performance improvement of 2.3%, 2%, 1.2%, 2.2%, 1.8%, 1.7%, 5% on CD, ESC-50, LC, NS-5h, SC-5h, FSD50k and VL tasks, respectively. That is a considerable improvement across the board on 7/10 tasks.
>
> At the same time, it's worth keeping in mind that MW-MAEs demonstrate better performance characteristics for different encoder/decoder complexities as well as different amounts of pre-training data used as shown in Sec 4.4.
>
> ---
>
> ## Questions
>
> > For instance, is it possible to make these window sizes trainable?
> >
>
> While plausible, learnable window sizes for MHA are a non-trivial challenge and at this stage are beyond the scope of this paper.
>
> > Alternatively, more engaging designs and comparisons could be conducted by examining the relationship between the number of attention heads, network hierarchies, and window sizes.
> >
>
> We have already presented ablations for different encoder and decoder complexities in Sec 4.4 in the main text of the paper. While more configurations can certainly be explored, encoder complexities, for the sake of comparison with previous works, have been limited to known ViT encoders.
>
> As for more comparisons for relationship between number of attention heads and window sizes, we have just presented them in the previous comment.
>
> ---
>
> ## Conclusion
>
> We thank the reviewer for engaging in a dialogue with us and for valuation suggestions. We hope we were able to address your concerns, and hope to hear back!
>
> ---
>
> ## References
>
> [a] Liu et al. 2021, “Swin transformer: Hierarchical vision transformer using shifted windows.”
> [b] Dong et al. 2022, “CSWin Transformer: A General Vision Transformer Backbone With Cross-Shaped Windows”.
> [c] Beltagy et al., “Longformer: The Long-Document Transformer.”
> [d] Fan et al. 2021, “Multiscale vision transformers".
> [e] Li et al. 2022, “Mvitv2: Improved multiscale vision transformers for classification and detection”.
> [f] Zhu and Omar, 2023, “Multiscale audio spectrogram transformer for efficient audio classification”.
> [g] Chen at al, 2022, “A hierarchical token-semantic audio transformer for sound classification and detection”.
> [h] Saeed et al., "Contrastive Learning of General-Purpose Audio Representations", ICASSP 2021.
> [i] Niizumi et al., "BYOL for Audio: Self-Supervised Learning for General-Purpose Audio Representation", IJCNN 2021.
> [j] https://hearbenchmark.com/hear-leaderboard.html

---

> ### Author Response · Authors · 2023-11-20
> **A Gentle nudge to reviewer M3PT**
>
> Dear reviewer M3PT,
>
> We have incorporated your comments and suggestions, as well as conducted additional experiments for window/attention head ablations. Given that the discussion period will end soon, we were hoping you could go through our response and tell us if it addresses your concerns, as well as answer any further questions you might have. We would highly appreciate it!
>
> Best,
> Authors.

---

> ### Author Response · Authors · 2023-11-23
> **Last few hours of the discussion period**
>
> Dear Reviewer M3PT,
>
> We have incorporated your suggestions, responded to your questions, and as per your recommendation, conducted additional full pre-training experiments to elucidate the effect of window size and attention heads on downstream performance.
>
> We thank you for your invaluable feedback! As the discussion period will end in a few hours, please take a look at our rebuttal and discourse with the other reviewers, and consider taking part in the discussion before the discussion period ends! We would greatly appreciate it.
>
> Best,
> Authors

---

### Author Response · Authors · 2023-11-19
**Overall response and overview of changes to the paper.**

We would like to thank the reviewers for taking the time to review our papers, for their questions and for their valuable feedback.
Here we summarize our responses to the reviewers, as well as the additions to the paper.

## Questions regarding performance improvements over standard MAEs
Reviewers M3PT and 4eUM raised concerns regarding performance improvements MW-MAE achieves over MAEs, citing that in some cases the performance improvements are marginal, and in some cases MAEs perform better.
As we have edified in our responses:

1. Throughout the paper, we have provided several experiments that clearly show that MW-MAEs always outperform MAEs in terms of overall score across the board, including extensive ablations covering scalability (encoder and decoder complexities), low-data performance, and adaptability to different patch sizes. The ViT-L based MW-MAE makes an absolute performance jump of 2.6% in overall score over its standard MAE counterpart, which is not a minuscule improvement.

2. At the task level, MW-MAE consistently performs better than MAE on 7/10 tasks. Eg. ViT-L-based MW-MAE from Table 2 achieves an absolute performance improvement of 2.3%, 2%, 1.2%, 2.2%, 1.8%, 1.7%, 5% on CD, ESC-50, LC, NS-5h, SC-5h, FSD50k and VL tasks, respectively. On the rest of the 3 tasks (BO, Mri-S, Mri-T), MW-MAE performs on par with MAE within the presented 95% confidence intervals. We believe that the performance on these tasks has saturated, as highlighted in our responses.

## Questions regarding applicability

Regarding questions of contribution and applicability, we have already done experiments that go beyond the modality (ImageNet, Appendix D). We have also done additional experiments on supervised ASR (now included in the Appendix). Although preliminary, these experiments show promise regarding the wide applicability of our approach.

## Additional Experiments

Based on feedback from the reviewers, we have done the following additional experiments:
1. Window size and Attention head ablations (in response to Reviewer M3PT).
2. Additional in-domain linear probe and supervised experiment for MW-MAE Encoder-only setting  (in response to Reviewer a4K8).
3. Additional experiment on a frame level task, MAESTRO 5hr: (in response to Reviewer 4eUM)
4. Evaluating AudioMAE in a directly comparable setting: MW-MAE still outperforms AudioMAE by a considerable margin (in response to Reviewer a4K8).

These results have been added to the paper.

## Miscellaneous fixes

- Grammatical errors (Reviewers a4K8, fDLB)
- Explicitly stating evaluated downstream tasks in the main body of the paper (Reviewer 4eUM)
- Fix and streamline repetitive references (Reviewer a4K8)
- Quantitative information about throughput (Reviewer 4eUM)
- Parameter explanations in Sec 3.1 (Reviewer a4K8)

---

## Conclusion

We hope these changes address the concerns of the reviewers, and we look forward to answering any further questions!

---

### Meta-Review · Area_Chair_kN5M · 2023-12-04

**Metareview:**

This paper introduces a Multi-Window Multi-Head Attention module that replaces the Multi-Head Attention module in the Masked AutoEncoder. The authors show that this new model surpasses the MAE in audio learning, as demonstrated by empirical results on 10 downstream audio tasks. Three out of four reviewers consistently rated the score marginally above the acceptance threshold. The authors addressed the concerns raised by the reviewer who gave the low score in their rebuttal, but there was no response from that reviewer.

**Justification For Why Not Higher Score:**

The paper’s contributions do not appear to be significant enough, and the methods proposed lack sufficient novelty for a higher score.

**Justification For Why Not Lower Score:**

The authors addressed the concerns raised by the reviewer who gave the low score in their rebuttal, but there was no response from that reviewer.

---

### Decision · Program_Chairs · 2024-01-16

Accept (poster)